# Tropical cyclone-blackout-heatwave compound hazard resilience in a changing climate

Kairui Feng [1], Min Ouyang [2] & Ning Lin [1]✉

Tropical cyclones (TCs) have caused extensive power outages. The impacts of TC-caused blackouts may worsen in the future as TCs and heatwaves intensify. Here we couple TC and heatwave projections and power outage and recovery process analysis to investigate how TC-blackout-heatwave compound hazard risk may vary in a changing climate, with Harris County, Texas as an example. We find that, under the high-emissions scenario RCP8.5, long-duration heatwaves following strong TCs may increase sharply. The expected percentage of Harris residents experiencing at least one longer-than-5-day TC-blackout-heatwave compound hazard in a 20-year period could increase dramatically by a factor of 23 (from 0.8% to 18.2%) over the 21st century. We also reveal that a moderate enhancement of the power distribution network can significantly mitigate the compound hazard risk. Thus, climate adaptation actions, such as strategically undergrounding distribution network and developing distributed energy sources, are urgently needed to improve coastal power system resilience.

## Main

Hurricane (generally called tropical cyclones or TCs) threaten 59.6 million people in the United States (2018)[1,2] and are important initiating causes of large-scale failures of power systems. In 2017, Hurricane Maria devastated Puerto Rico's power grid, resulting in a power loss of 3.4 billion customer hours and the worst blackout in U.S. history[3]. Hurricane Irma (2017) deprived over 7 million customers of electricity, 2.1 million of whom still lacked access to electricity after four days[3]. Hurricane Harvey (2017), making landfall on the Texas coast, disrupted hundreds of overhead electricity lines and disabled over 10,000 megawatts (MW) of electricity generation capacity; local utility companies spent over a week to restore the system[4]. In 2020, Hurricanes Isaias, Laura, Sally, Delta, and Zeta caused ~3.8, 1.5, 0.9, 0.8, and 2.0 million power outages, separately[5]. Similarly, Hurricane Florence (2018) cut electric power for around 1.4 million customers; the system took two weeks to recover[6]. Hurricane Sandy (2012) affected over 8.5 million customers[7,8]. These disruptions, leaving millions of customers without electricity for days, call for an investigation of power system

resilience and ultimately a redesign of the energy infrastructure[9]. Moreover, the U.S. power grid may become more vulnerable to weather and climate-induced failures in the future due to climate change[10]. In particular, hurricane-induced power outages are likely to become more severe, as increasing evidence shows that hurricane intensity will increase due to climate change (e.g., refs. [11–17]). This potential change should be quantified and accounted for in planning future energy infrastructure.

Projected with greater confidence, climate change may also induce more heat extremes that are beyond human tolerance[18–21]. Heatwaves are the primary cause of weather-related mortality (due to heat cramps, syncope, exhaustion, stroke, etc.) in the United States[22], and they may harm mental health[23,24], sleep quality[25], and social stability[26] in different ways. Reference [27] is an early research to connect TCs with heatwave impacts. With TC-induced power outages, the impact of heatwaves would dramatically increase, as air conditioning, with around 1.6 billion units in operation over the United States[28], is critical in reducing the vulnerability to extreme heat (the fatality risk under heatwaves without air conditioning is estimated to triple, as

[1]Civil and Environmental Engineering, Princeton University, Princeton, NJ, USA. [2]School of Artificial Intelligence and Automation, Huazhong University of Science and Technology, Wuhan, China. ✉e-mail: nlin@princeton.edu

revealed by a post-heatwave analysis in Chicago[29]). Reference 27 found that TC-heatwave compound events have been rare and affected only about 1000 people worldwide, as the seasonal peak of extreme heat precedes that of major TCs. Due to global warming, ref. 27 projected that the TC-heatwave compound hazard would affect a much larger population in the future, e.g., over two million in a world 2 °C warmer than pre-industrial times. However, ref. 27's projection neglected potential changes in TC climatology in the future, which may result in an underestimation of the compound hazard, given that TCs will likely become stronger (e.g., refs. 11–16) and possibly occur earlier in the season[30]. Also, ref. 27 focused on TC-heat compound hazard, but the potential impact of the compound hazard depends also on the reliability and resilience of the power system.

To better quantify the evolving joint impact of TCs and heatwaves, here we model the TC-blackout-heatwave compound hazard risk and resilience in a changing climate. We combine statistical-deterministically-downscaled TC climatology and directly-projected heatwave climatology from general circulation models (GCMs), and we explicitly model the power system failure and recovery process under the climate hazard scenarios. We investigate how the risk of residents experiencing prolonged TC-blackout-heatwave compound hazards may change from the current to future climates. To illustrate, we apply the analysis to Harris County (including a major part of Houston City) in Texas (see Methods and Supplementary Fig. S1 for a description of the area's geography and power network). Harris County has the highest population density along the Gulf Coast and, as located in the subtropics, may face a disproportionally large increase in heatwaves[31] and TCs[32] in a warming climate.

As TCs cannot be well resolved in typical GCMs due to their relatively small scales, we apply large datasets of synthetic storms generated by a deterministic-statistical TC model[33] for the study area[34], for the historical climate of 1981–2000 based on the National Centers for Environmental Prediction (NCEP) reanalysis and for the future climate of 2081–2100 under the emissions scenario RCP8.5 based on six GCMs in the fifth Coupled Model Inter-comparison Project (CMIP5; see Methods). Based on the performance of the GCMs in terms of their historical simulations compared to the reanalysis-based simulation, we bias-correct the GCM-simulated storm frequency and landfall intensity distribution and combine the six simulations into a single projection for the future climate. Then we stochastically resample the synthetic storms from the combined projection to form 10,000 20-year simulations for the historical climate and 10,000 20-year simulations for the future climate (Methods). Here we focus on wind effects on the power grid (see Discussion) and apply a parametric model to estimate the spatial-temporal wind field for each synthetic storm. We apply a physics-based power outage and recovery model[35], validated by two historical cases (Hurricanes Harvey in 2017 and Ike in 2008) for the study area, to simulate the wind-induced power system failure and recovery process at a census-tract level for each synthetic storm (see Methods).

The U.S. National Weather Service issues warnings when a forecast heat index (HI) characterizing humid heat exceeds 40.6 °C. Reference 27 defines a compound TC-heat event as a major TC followed within 30 days by an HI greater than 40.6 °C at the site of landfall. Here we also define a heat event as an HI over 40.6 °C, but we are interested in a range of time scales, especially over 5 days following the TC landfall. (Considering that TCs usually start to interrupt the life pace of local residents at least 2 days ahead of landfall[36], a greater-than-5-day power outage plus heatwave means affected residents cannot resume normal life for over a week, which is usually a benchmark for resilience design for critical infrastructure systems[37]). Based on the same reanalysis and GCM datasets, we calculate and bias-correct the HI for the study area during and after the landfall of each synthetic storm (see Methods). In addition, we modify the HI to statistically account for the interdependence between TCs and

heatwaves (Methods). Combining the obtained HI dataset and TC power outage analysis enables us to estimate the risk of Harris residents experiencing prolonged (e.g., 5-day) heatwaves (HI >40.6 °C) after losing power due to TC-induced damage. We first focus on the estimated risk for the high-emissions scenario RCP8.5 and the end-of-the-century (2081–2020) time frame (Results); we then investigate the sensitivity of the estimated risk to the emission scenario, the mid-of-the-century time frame, and, considering the large uncertainty regarding how TC frequency will change (e.g., refs. 15, 16, 38, 39), the storm frequency projection (Discussion). Also, we assume the power system and recovery operation will remain the same in the future (Results), but we investigate the possible effects of heatwaves on power restoration and future power capacity upgrade (Discussion).

In addition to quantifying the risk, we explore efficient strategies to enhance power system resilience for combating future TC-blackout-heatwave compound hazards. Consistent with previous results[8], our network analysis (see Methods) shows that local power failures have a disproportionally large non-local impact on the power system; the reliability of local power distribution networks (i.e., the final stage of energy infrastructure) is particularly critical. Thus, we propose an undergrounding strategy (burying parts of the power network covered by anti-water pipelines[40–42]; see Methods) to protect a small portion of wires that are close to the root nodes of local power distribution networks. We show that such a targeted undergrounding strategy is much more efficient than the widely-applied uniform undergrounding strategy in increasing the resilience of the power system and decreasing the risk of future TC-blackout-heatwave compound hazards.

## Results

### Historical cases

Hurricanes Harvey and Ike are the main events in the past 20 years that led to significant power outages (over 10% of residents lost power) in Harris County. Ike had a higher gust wind observation (~92 mph) when it hit Houston; Harvey was weaker (<50 mph) but lasted longer and brought heavy rainfall[43]. The wind during Ike broke many more poles, leading to a larger power outage. As shown in Fig. 1 for the total impact on Harris County from Hurricanes Ike and Harvey, results from the power grid outage and recovery model compare relatively well with the observation (although the model slightly overestimates the initial power outage under Harvey), indicating the model's success in capturing both large and relatively small power outage events. The model estimates a relatively small power outage (1% [0–1%]) after the 5-day

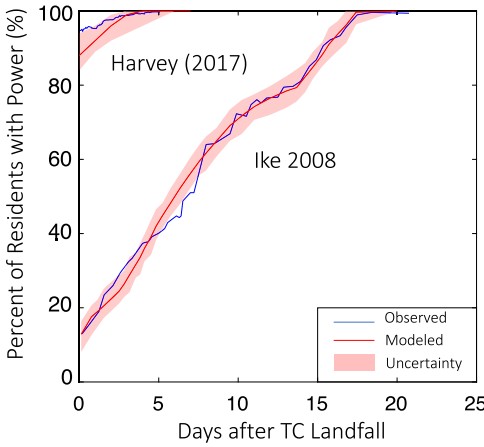

**Fig. 1 | Comparison of the simulated and observed power outage and recovery process for Hurricanes Harvey and Ike in Harris County, TX.** Red curve shows median values of the simulation results, with the 32 to 68% quantile range shown by shade. Blue curves show the observations of the power outage.

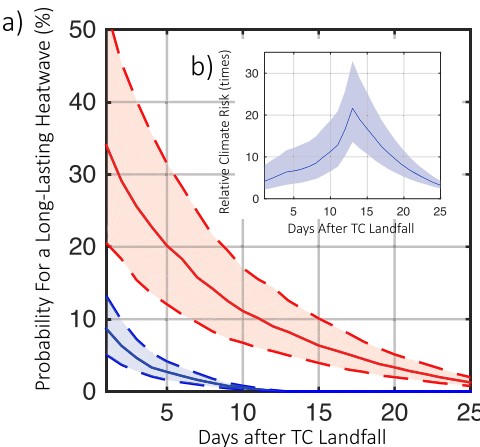

**Fig. 2 | Analysis of likelihood and duration of post-TC heatwaves for Harris County. a** Probability for a post-TC heatwave lasting over a certain duration in the historical climate (blue curve showing median, with 32 to 68% quantile range shown by shade) and future climate (red curve showing median, with 32 to 68% quantile range shown by shade). **b** Relative climate risk, defined as the ratio of future probability and historical probability of post-TC heatwave, as a function of heatwave duration (shade shows 32 to 68% quantile range).

restoration period for Harvey and a large power outage (63% [60–66%]) for Ike for over 5 days, agreeing with observations (Fig. 1). The model results at the census tract level for Hurricanes Ike and Harvey compare also relatively well with observations (Supplementary Fig. S2), with a mean error over all census tracts less than 10%.

## Post-TC heatwaves

While heatwaves did not follow Hurricanes Ike and Harvey, they may become more likely to follow future TCs. Under RCP8.5, the projected change in global mean surface air temperature for the late 21st century relative to the reference period of 1986–2005 is 3.7 °C based on all GCMs in the IPCC report[44]. The six GCMs used in this study predict an increase of the average global temperature by 3.4 °C in the future climate (2081–2100) compared to that in the historical climate (1981–2000). We analyze the synthetic TC and heatwave datasets to investigate the likelihood and duration of future post-TC heatwaves. Figure 2a shows how the probability of experiencing a heatwave with a certain duration following a TC changes from the historical to future climate for the study area. The probability of a heatwave, especially a long-lasting one, following a TC is quite small for the historical climate, which is consistent with the observation that TC-heatwave hazards have affected only ~1000 people worldwide during 1979–2017[27]. However, the probability curve is much higher for the future climate. The probability of a post-TC heatwave lasting over 5 days is 2.7% (1.8–4.2%) under the historical climate but 20.2% (12.0–31.5%) in the future climate. For a post-TC heatwave to last over 12 days is almost impossible in the historical climate, but a nonnegligible probability of 7.5% (4.8–11.9%) exists for it to happen in the future climate. To better reveal the time scale of climate change impact, Fig. 2b shows the relative climate risk, defined as the probability of experiencing a heatwave lasting for a certain duration following a TC in the future climate divided by that in the historical climate. The relative climate risk increases sharply with the duration, reaching the peak around 13 days. The probability for a 1-day heatwave following a TC in the future climate would be ~5 (3–9) times larger than that in the historical climate, the probability for a 5-day heatwave would be 7 (4–12) times larger, and the probability for a 13-day heatwave would be 22 (14–33) times larger. This time scale pattern of climate change impact should be considered when developing maintenance and emergency response strategies for urban infrastructure systems. For example, the

resilience criteria of these systems should be enhanced to avoid the "resonance" effects of climate change. A typical recovery cycle for the power system is currently 5 days or longer[37]; the dramatic change in the climate risk may render such a time scale of recovery not resilient.

## Blackout and compound hazard risk

Incorporating the power outage and recovery modeling, we analyze the TC-induced blackout and TC-blackout-heatwave compound hazard risk for the study area. We apply an agent-based approach and record the largest duration of the hazards each resident could experience during each of the 20,000 20-year synthetic simulations. Figure 3a, b show the estimated percentage of Harris residents who may not experience post-TC power outages longer than a certain duration over a 20-year period under the historical and future climates, respectively. On average, the TC-induced power outage could affect 50% of the residents during a 20-year period in the historical climate and 73% of the residents in the future. 12.8% of the residents would not be subject to any post-TC blackout during the 20-year period under the worst case in the historical climate, while only 2.7% of the residents would not be affected under the worst case in the future climate. In the historical climate over a 20-year period, on average, 14% of Harris residents could face at least one longer-than-5-day post-TC power outage, which is less than one-third of the expected level of 44% in the future. For 90% of the cases under the historical climate, the utility company (CenterPoint Energy) could fully repair the power system within about 15 days, which matches recent-year records (e.g., 12 days for Hurricane Irma in 2017, and 13 days for Hurricanes Michael and Florence in 2018[5]). With the same response strategies and resources, the utility company might spend over 25 days in repairing the power system in the future under severe TCs. By comparison (with Fig. 1), the probability of experiencing the scale and duration of power outage induced by Hurricane Ike during a 20-year period is about 10% in the historical climate and 35% in the future. Hurricane Harvey is a below-average event in both the historical (87%) and future (96%) climates. These analyses show that climate change may dramatically increase the outage risk of the power system, especially the tail of the risk that people will face.

Figure 3c, d show the estimated percentage of Harris residents who may not experience any TC-blackout-heatwave compound hazard longer than a certain duration over a 20-year period under the historical and future climates, respectively. The chance for a resident to experience at least one longer-than-5-day compound hazard is almost zero (0.8%) under the historical climate. However, the chance for a resident to be affected by at least one such compound hazard in the future climate (18.2%) would be 23 times larger. Recall that, due to climate change, the 5-day post-TC power outage risk is estimated to increase by 3 times (Fig. 3a, b), and the 5-day post-TC heatwave compound hazard is estimated to increase by seven times (Fig. 2b). The 23-time increase of TC-blackout-heatwave compound hazard risk, which is larger than the product of the two factors as if TCs and heatwaves are climatologically uncorrelated (21 times), indicates that strong TCs and long-lasting heatwaves may become more likely to co-occur under climate change. In other words, strong TCs leading to larger power outages may be more likely followed by longer-lasting heatwaves, affecting over 18% of Harris residents towards the end-of-the-century.

To investigate the spatial distribution of the risk, we estimate the percentage of residents to experience at least one longer-than-5-day TC-induced power outage (Fig. 4a, b) and TC-blackout-heatwave compound hazard (Fig. 4c, d) over a 20-year period for each census tract in Harris County in the historical and future climates. Changing from the historical to the future climate, the power outage risk of all the census tracts would at least double. The compound hazard risk would increase even more dramatically. Over 95% of people in all census tracts would experience no longer-than-5-day compound

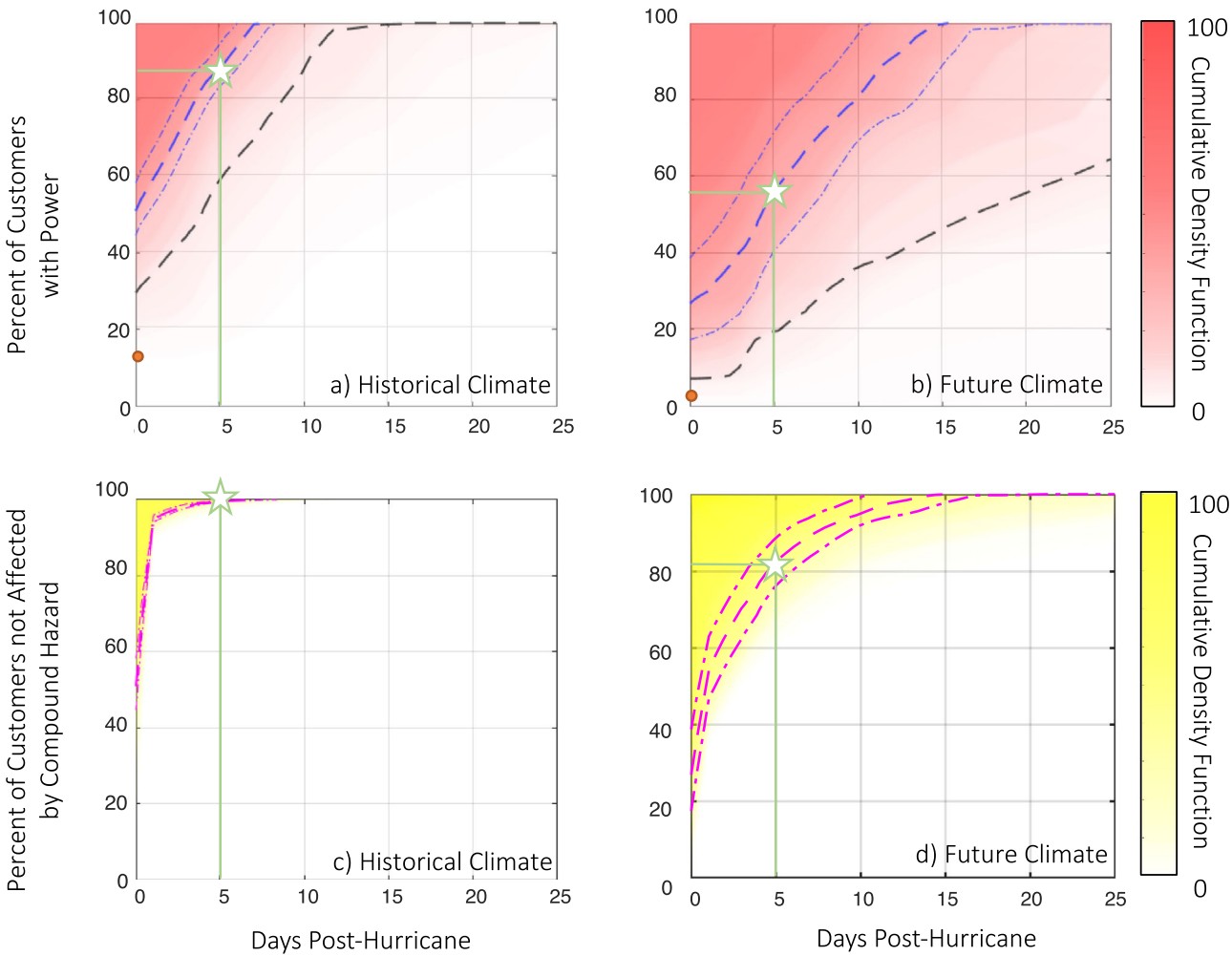

**Fig. 3 | Analysis of TC-heatwave compound hazard risk for Harris County.** The estimated percentage of residents who may not experience outage days after TC landfall in a 20-year period are shown for **a** the historical climate and **b** the future climate. The orange dot shows the simulated worst-case power outage over all simulations. The dashed black line shows the 10% worst case of power recovery. The estimated percentage of residents who may not experience TC-blackout-heatwave compound hazard days after TC landfall in a 20-year period are shown for **c** the historical climate and **d** the future climate. In all panels, the dashed and dashed-dotted curves show expectation and ±1σ range based on all simulations. The shade shows CDF, indicating the probability of less than a certain (y) percentage of residents not affected, with a darker color corresponding to higher probability and thus higher risk. Green stars highlight the expected percentage of residents not affected 5 days post-TC landfall.

hazard in the historical climate; in the future, over 95% of census tracts would have over 5% of the residents experiencing at least one such compound hazard during a 20-year time period. The distribution of risk is heterogeneous, which implies a heterogeneous distribution of the resilience level of the power system, as the heatwave is constant (given the GCM resolution) and TC winds vary only slightly over this relatively small region. The spatial pattern shows that residents near the center of Houston (the middle and lower part of the County) may experience lower power outage and compound hazard risk than residents in rural (e.g., the upper part) areas of the County. The varying densities of power substations and spatial patterns of distribution networks induce most of this difference.

**Scaling law of power system failure**

To better understand the power system, we apply network analysis to investigate links between local disruptions (the failure of a specific pole or distribution/transmission line) and global failures, using our large datasets of synthetic storm events. For each storm event, we calculate $W(x)$, the percentage of service interruptions induced by all local disruptions that affect more than x customers, and $P(x)$, the percentage of local disruptions that affect more than x customers.

$W(x)$ also represents the probability for a customer to be subject to a disruption that affects more than x customers, and $P(x)$ is the probability for a disruption to affect more than x customers. Figure 5a shows the obtained generalized scaling law[8] between $W(x)$ and $P(x)$, for each simulated event and averaged over all the events for the historical climate. Rather than a linear relationship indicating a uniform distribution of damages, the concave scaling curve shows that, on average, the largest 20% of local disruptions are responsible for 72% (71–75%) of the global power outage (measured by the number of affected customers). This result obtained from the synthetic analysis for Harris County is comparable with a previous empirical result for Upstate New York that during Hurricane Sandy the top 20% of local disruptions induced 79–89% of the service interruptions[7]. In fact, the difference in the scaling curves among the full range of synthetic TCs is quite small, which confirms the robustness and generalizability of the scaling law in describing a TC-damaged power system. Further, ref. 8 found that the scaling curve developed using data on failures in daily operation is also similar, suggesting that the power network vulnerability (i.e., local disruptions inducing large-scale interruptions) exists independently of exogenous effects. Thus, reliability-based redesign against various TCs may not need to differ

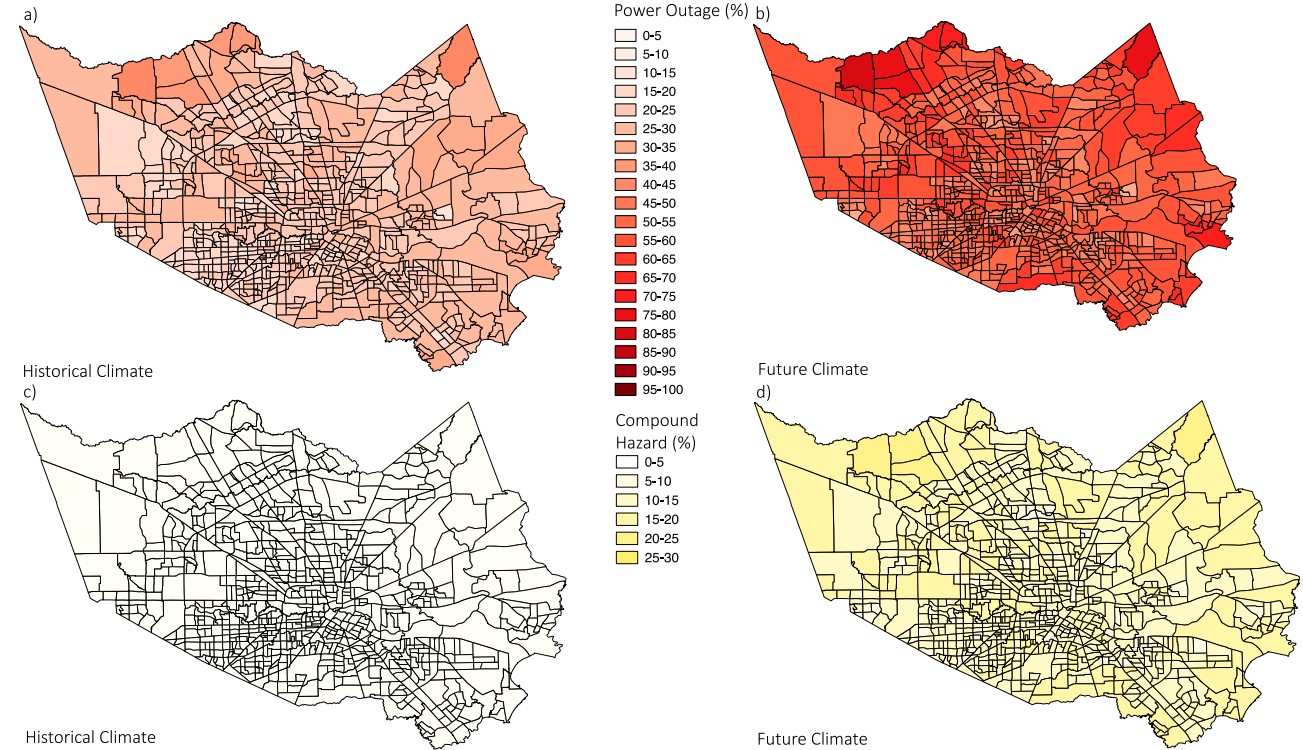

**Fig. 4 | Spatial distribution of estimated blackout and compound hazard impact.** Estimated percentage of residents facing at least one longer-than-5-day post-TC power outage (**a**, **b**) and blackout-heatwave compound hazard (**c**, **d**) in the historical (**a**, **c**) and future climate (**b**, **d**) for each census tract in Harris County.

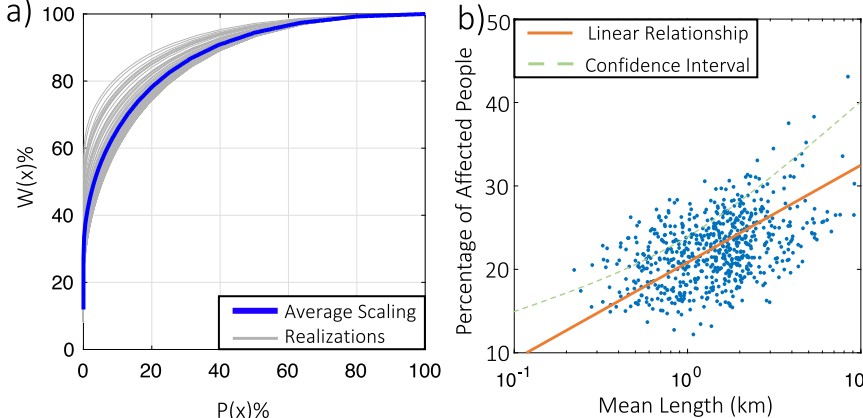

**Fig. 5 | Network analysis of the synthetic TC-induced power outage in Harris County. a** Generalized scaling law for the Harris power system: empirical probability W(x) of a customer being affected by a disruption that affects more than x customers vs. empirical probability P(x) for a disruption to affect more than x customers during a storm event. The blue curve shows the average overall synthetic events in the historical simulation; the gray curves show randomly selected 100 events. **b** Percentage of residents experiencing a longer-than-5-day power outage averaged over all synthetic storms simulated for the historical climate vs. harmonic mean length (reciprocal of the arithmetic mean of reciprocals) of power distribution network sectors for each census tract in Harris County. The red line shows the linear fit; dashed curves show the ±1σ uncertainty range.

fundamentally from the daily-operation-based enhancement, as confirmed later (Fig. 6).

Large power outages are usually induced by local damages (as shown in our analysis and previous studies[8]), and large portions of local damages are due to distribution network failures, so we analyze the connection between power outages and local distribution network topology. Figure 5b shows the correlation between the percentage of residents experiencing a longer-than-5-day power outage (averaged over all simulated synthetic storms for the historical climate) and the harmonic mean length of the power distribution network sectors at the census tract level. The results show that the longer the length of the local power distribution networks, the higher

the risk. The mean length of the power distribution networks is highly related to the pattern of urban development. A region with a lower population density may have a larger mean length of power distribution networks, as it may have fewer substations to support residents who live relatively far away. Thus, our results indicate that, for TC-prone regions, the scaling of urban development may have contributed significantly to the spatial distribution of power outage risks. The observed high correlation between the power outage rate and the mean length of distribution networks and the generalized scaling relationship between the probability of local failure and the failure impact, shown in Fig. 5, can be explained theoretically and generally for acyclic (the most common) power distribution

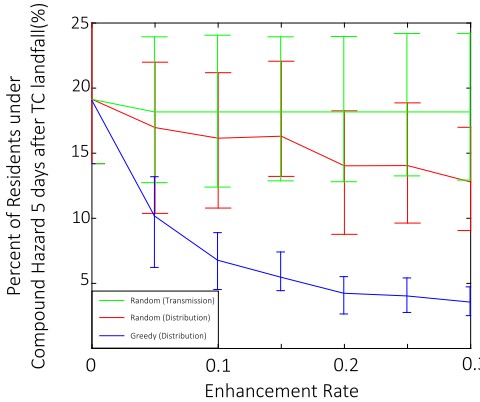

**Fig. 6 | Reduced impact of longer-than-5-day TC-blackout-heatwave compound hazard under various risk mitigation strategies.** Solid curves show the percentage of affected residents in a 20-year period in the future as a function of network enhancement rate with three strategies: randomly undergrounding power transmission networks (green), randomly undergrounding power distribution networks (red), and greedily undergrounding power distribution networks (blue). Curves show the expectation; error bars show the ±1σ uncertainty range.

networks (see theoretical analysis and Fig. S3 in Supplementary Materials).

## Efficient undergrounding strategy

Based on an improved understanding of the power network vulnerability, we test three network enhancement strategies to mitigate the TC-blackout-heatwave compound hazard risk in the future. Figure 6 compares the reduction of the TC-blackout-heatwave compound hazard risk, measured by the percentage of residents affected by longer-than-5-day compound hazard, as a function of the enhancement rate for the three strategies; the enhancement rate is the total length of enhanced networks divided by the total length of the network sections that could be enhanced. Random enhancement of the high-voltage transmission network (burying transmission branches randomly) provides limited improvement of the global system performance. This finding confirms that the local distribution networks dominate the global pattern of the power outage and subsequent recovery process. Randomly undergrounding low-voltage power distribution networks improves the system performance linearly, which means that the power distribution networks without protection face the same risk as before the enhancement. Given the acyclic topology of the power distribution networks, a greedy undergrounding strategy is used to protect a small portion of wires close to the root nodes of the distribution networks. In the simulation, the algorithm simply protects the root sector of the longest overhead branches of the distribution network iteratively until the enhancement rate is reached (i.e., "greedy" undergrounding). The greedy undergrounding strategy improves the system performance much more than the other two strategies. For example, with the first 5% power distribution networks undergrounded, the expected percentage of residents who might face the compound hazard for over 5 days drops to 11.3% from 18.2% (Fig. 3d). This performance improvement (6.9%) is ~15 times more than that of randomly undergrounding distribution networks (0.5%) or randomly undergrounding transmission network (0.1%), demonstrating a superior cost-efficiency of the greedy strategy. The topological law for power outages that we found here may help utility companies to plan resilient power networks to combat the changing climate.

## Discussion

The results of this analysis demonstrate how dramatically the impact of TCs may increase over time, due to compound effects of storm and heatwave climatology change. For Harris County, the expected

(average) percentage of residents experiencing at least one longer-than-5-day heatwave without power post-TCs in a 20-year period would increase from 0.8% in the historical climate (1980–1999) to 18.2% towards the end of the 21st century (2080–2099), under the high-emissions scenario RCP8.5. For the current (2000–2019), near-future (2020–2039), and mid-of-the century (2040–2059) time frames, which are also highly relevant to decision making, this impact percentage would increase to 2.2, 5.1, and 6.7%, respectively (see Supplementary Fig. S4). Even if we account for the uncertainty in the prediction of storm frequency[16] and remove the predicted increase in the storm frequency for the study area (by applying the frequency in the historical climate), the impact percentage would still increase significantly, to 11.2% towards the end of the 21st century (Supplementary Fig. S5). Only if we assume that the warming is well controlled under the low-emissions scenario RCP2.6 and the TC activity (including both frequency and intensity) remains the same as the historical level, the compound hazard would change slightly, with the impact percentage changing from 0.8% in the historical climate to 1.0% towards the end-of-the-century (Supplementary Fig. S6). If the RCP8.5 scenario is considered an upper bound[45] and the ideal RCP2.6 scenario a possible lower bound, this additional analysis indicates that the compound hazard risk may be largely avoided under rigorous climate mitigation policies in line with the Paris Agreement. However, large uncertainties exist in the economic and political systems[45] as well as in the climate system (including possibly complex physical interactions between TCs and heatwaves); climate adaptation and risk mitigation are still necessary and urgent.

Other uncertainties exist in the modeling of the future compound hazard and risk. As the first attempt in quantifying the compound hazard risk, here we focus on the dominant power damage effects—winds and induced falling trees[46]. Located relatively high above the sea level (see elevation map in Supplementary Fig. S1), our study area, Harris County, is affected mainly by extreme winds, as evident in historical events including Hurricanes Ike and Harvey (though lower Houston beyond the Harris County was heavily flooded by Harvey's extreme rainfalls and compound flooding). Accounting for significantly less impact (<10% TC-induced power outages[46]), however, flooding and associated debris during TCs can also cause damage to the power system, and they usually impede the early recovery of a power network. Flood-prone regions may experience higher risks than estimated here, and future development of the modeling method must take flood impact into account. A precise prediction of flood-induced power outage and recovery requires an accurate prediction of the magnitude and timing of the flooding (from storm surge and/or heavy rainfall)[47], power system vulnerability under flooding[48], and operational logistics of local utilities when repairing wetted power system components. The heatwave may also trigger power outages due to excessive power demand; however, these outages are usually restored at a time scale of hours[49], rather than at a daily or weekly restore time scale for the TC damage-induced outages that we consider in this study.

The power system operation may also be affected by extreme heat. According to the Occupational Safety and Health Administration (OSHA) criteria, outdoor workers can work for only limited hours (<75% of normal hours) under extreme heat and humidity (HI >39.4 °C). Assuming there is no advanced technology to improve outdoor working condition and the recovery operation follow the OSHA criteria, the expected percentage of Harris County residents experiencing at least one longer-than-5-day TC-blackout-heatwave compound hazard in a 20-year period under the future climate could increase from 18.2% (workers working normally) to 23.3% (see Supplementary Fig. S7). Aging of the power system, which is not accounted for, can also reduce the resilience and increase the compound hazard risk, although proactive system maintenance may reduce this effect. Moreover, the scaling relationship between the power network and the

population in a growing city may enlarge the affected population in the future, as power facilities usually develop much slower than the growth of the local population (a 0.87-power scaling[50]). Given this scaling effect, unbalances between the local population and network density may increase, and thus future risks may be higher than those estimated herein.

On the other hand, we do not account for the benefits of backup generators or solar panels here. These local devices could temporarily support residents who lose power from the main grid—another possible way to mitigate the impacts of TC-blackout-heatwave compound hazards. The power demand and dependence may reduce during TCs thanks to evacuation, but high power demand may still exist if electrical vehicles are increasingly adopted and used for evacuation[51]. Also, as the temperature increases in the future climate, it is possible that utilities will have the incentive to upgrade the power system capacity to match the higher temperature-related power demand. However, improving the power capacity would not significantly improve the power system reliability and resilience under TCs, even when the capacity is increased by 50% (see Supplementary Fig. S8).

Strategically undergrounding local distribution networks can efficiently enhance the resilience of the power system to adapt to climate change. Our analysis shows that undergrounding only 5% of the distribution networks close to the root nodes can reduce the expected percentage of Harris County residents experiencing at least one longer-than-5-day TC-blackout-heatwave compound hazard from 18.2% to 11.3%. As the power outages under TCs and daily operation are both dominated by the generalized scaling law (e.g., the top 20% of local damages would trigger over 80% of total outage), the reliability-based enhancement of power grids against TCs can be considered jointly with daily-operation-based enhancement. This finding points further to the potential to develop a unified design framework for enhancing the power system resilience against various damage sources. The potential co-benefit and improved cost-efficiency induced by the unified strategy may better motivate utility companies to mitigate the compound hazard risk. Furthermore, the power outage and compound hazard we consider herein can significantly disrupt local business and supply chains, leading to secondary losses[52], and the enhanced connectivity of local and global economics potentially would further foster the impact[53]. The coupled modeling of the compound hazard and induced economic disruption may be applied in future studies to quantify the cost-benefit[54] of proposed risk mitigation strategies.

To develop efficient and economic risk mitigation strategies, quantifying the reliability and resilience of infrastructure systems under the impact of future compound hazards is essential but is still associated with various uncertainties, as discussed above. Risk analysis requires continuous updates wherein improved modeling and new data become available. As extreme climate events become more frequent, coastal megacities also develop rapidly[1,55]. To ensure sustainable development, effective strategies to mitigate the TC-blackout-heatwave compound hazard risk, a pronounced example of physical-social connected extremes[56], should be carefully developed. The analysis results for Harris County obtained in this study may be qualitatively representative of coastal megacities in the subtropical regions of the United States, but the developed methodological framework can be applied to quantify power system resilience and develop risk mitigation strategies for any TC-prone areas. More generally, this study demonstrates an interdisciplinary approach that integrates the state-of-art climate science and infrastructure engineering to project climate change impact and develop risk mitigation strategies, with the goal of achieving resilient and sustainable communities.

## Methods

### TC projection

TCs cannot be well resolved in typical climate models due to their relatively small scales, except perhaps in a few recently developed

high-resolution climate models (e.g., ref. 38). Dynamic downscaling methods can be used to better resolve TCs in climate-model projections (e.g., ref. 57), but most of these methods are still computationally too expensive to be directly applied to risk analysis. An effective approach is to generate large numbers of synthetic TCs under reanalysis or GCM-projected climate conditions to drive hazard modeling. In this study, we apply large datasets of synthetic storms generated by a deterministic-statistical TC model[33]. This model uses thermodynamic and kinematic statistics of the atmosphere and ocean derived from reanalysis or a climate-model to produce synthetic TCs. The model, when driven by a reanalysis environment, is shown to generate synthetic storms that are in statistical agreement with observations[58], and it has been widely used to study TC wind, storm surge, and rainfall hazards under climate change effects (e.g., refs. 32, 33, 59).

Specifically, we apply the synthetic TC datasets generated with this model by ref. 34 for the Houston area. Each synthetic storm passes within 300 km of Houston, with a maximum wind speed of at least 22 m/s (sensitivity analysis shows that the hazard modeling results are not sensitive to the storm selection radius when it is greater than 200 km). The datasets include 2000 synthetic TCs under the historical climate over the period of 1981–2000 based on the National Centers for Environmental Prediction (NCEP) reanalysis, as well as 2000 synthetic TCs for the historical climate (1981–2000) and 2000 synthetic TCs for the future climate (2081–2100 under the high-emissions scenario RCP8.5) based on each of six CMIP5 GCMs (chosen based on data availability and following previous studies): the National Center for Atmospheric Research CCSM4, the United Kingdom Meteorological Office Hadley Center HadGEM2-ES, the Institute Pierre Simon Laplace CM5A-LR, the Japan Agency for Marine-Earth Science and Technology MIROC-5, GFDL-CM3.0 (NOAA Geophysical Fluid Dynamics Laboratory), and the Japan Meteorological Institute MRI-CGCM3.

To account for possible biases in the climate projection, we bias-correct storm frequency and landfall intensity and apply stochastic modeling to resample the storms. Specifically, for each GCM-driven projection, we bias-correct the projected storm frequency for the future climate by multiplying it by the ratio of the NCEP estimated frequency (where the NCEP frequency was calibrated to be 1.5 times/year for Houston using historical data[60]) and GCM-estimated frequency for the historical climate, assuming no change in the model bias over the projection period, following ref. 32. We apply the same assumption to bias-correct the projected landfall intensity (maximum wind speed) for the future climate, through cumulative density function (CDF) quantile mapping based on a comparison of the NCEP and GCM-estimated CDF of the annual maximum landfall intensity for the historical climate, similar to ref. 61, and reweighting each TC simulation based on the bias-corrected intensity distribution. To obtain a single projection for the future climate, we combine the bias-corrected projections from the six GCMs, weighted (as in ref. 62) according to their performance in estimating the CDF of the annual maximum landfall intensity for the historical climate compared to the NCEP estimates. Finally, we stochastically resample the storms from the NCEP historical analysis and combined projection for the future climate; 10,000 20-year simulations are generated for the historical climate and 10,000 20-year simulations are generated for the future climate. The storm occurrence times (generated in the physical model according to storm climatology) are matched with the heatwave analysis for the study area. For each sampled storm, we generate the spatial-temporal wind field, employing the classical Holland wind profile[63], accounting for the effects of surface friction and large-scale background wind based on ref. 64, and converting 1-min. mean winds to 3-s wind gusts using gust factors[65], to drive the power grid outage analysis.

**Heatwave projection.** Similar to ref. 27, the HI is calculated at the daily level as a function of near-surface (at 2 m) air temperature (daily

maximum), specific humidity (daily mean), and surface pressure (daily mean). To be consistent with the TC simulation, we obtain these data for Houston from the NCEP reanalysis and 6 GCMs mentioned above, matched in date with the landfall (defined as when the storm is at its closest point to Houston) of each generated synthetic storm. The GCM-projected future HI is bias-corrected[31,66] by adding to it the difference between the NCEP reanalysis and GCM-estimated historical HI (monthly average cubic-spline interpolated to daily). When combining the datasets of storm and heatwave events (HI >40.6 °C), we account for their possible correlation. Based on observations, ref. 27 found that TCs arrive after an anomalously high HI from amplified air temperatures and specific humidity; after TC passage, HI anomalies decrease to negative and return to zero within ~10 days. Reference 27 neglected the interdependence between storms and heatwaves when estimating compound TC-heat events on a 30-day time scale. Here, concerning compound events on shorter time scales, we account for the interdependence statistically: we add the composite impact of TC passage to the meteorological variables used to calculate the HI, where the composite impact is estimated based on historical data (Fig. 3a in ref. 27). Accounting for this correction reduces the HI values within 5 days of TC passage and thus the probability of 5-day compound TC-blackout-heatwave events defined in this study. Also, we consider hazard compounding as when the heatwave starts before or on the day of TC landfall (i.e., neglecting heatwaves that occur after landfalls), which results in a conservative estimation of the compound risk.

**Power system modeling.** While various statistical models[67,68] have been developed to estimate TC-induced blackout, we employ a physics-based model to better account for future evolving factors, e.g., climate change, infrastructure upgrade, and utility maintenance. Specifically, we apply the power grid outage and recovery model developed by ref. 35 to simulate TC impact on the electric power system in Harris County, TX. This system serves approximately 1.7 million residents in a service area of around 4600 km², and over 90% of metropolitan households in Southwestern America use air-conditioning[69]. The power grid includes high-voltage transmission networks, where 551 transmission lines connect 23 power plants and 394 substations, and low-voltage generated star-like distribution networks (discussed below), which contain ~40,000 branches[35] (Supplementary Fig. S1).

Given the TC wind hazard (i.e., local maximum wind gust during the storm), the power grid failure model first applies probabilistic fragility functions to estimate the damage states of five main vulnerable component types of the power network: transmission substations, transmission lines, distribution nodes, distribution lines, and local distribution circuits. Component failures alter the power grid topology and may separate the power grid into unconnected subgrids. A direct current (DC)-based power flow simulation is then performed to capture the power flow pattern in each sub-grid, and the local demand is cut when an overflow happens until the system achieves a steady state (refs. 70, 71 used a similar approach). The power system is open and connects with systems outside the study area via transmission lines; the performance of the power grid outside the study area is assumed normal. The recovery model, developed based on emergency response plans and operational data, applies estimated recovery resources based on a priority-oriented strategy to repair damaged transmission substations, transmission lines, and critical facilities vital to public safety, health, and welfare before local distribution networks[35]. The power grid outage and recovery model was calibrated for the study area by ref. 35 using observed data for Hurricane Ike (2008). We further evaluate the model by adding data from Hurricane Harvey (2017). The same wind field modeling method applied to the synthetic storms is used for these two historical storms with storm characteristics (i.e., track, intensity, and size) taken from the extended best track data[72].

The power system model we use here, though physics-based, cannot resolve all details during a power outage and recovery process. Given that distribution networks and protective devices data are generally not available, star-like network[35] and minimal spanning tree (MST)[73] models (and models combining these two[73,74]) are usually used to generate synthetic distribution networks, and features that may have minor effects on predicting the daily-scale power outage and recovery under hurricanes including protective devices are usually neglected[73–75]. Based on these assumptions, the adopted simulation framework is arguably the best model that we can use to capture the power outage and recovery process at a mesoscale level, e.g., for each zip code or census tract, rather than at the individual household level, for risk and resilience analysis. The model using star-like distribution networks and neglecting protective devices is also validated for Hurricane Ike and Harvey at both the county level (Fig. 2) and census tract level (Fig. S2). The well consistency between the daily scale simulated results using this model and the real observations, together with the fact that this research mainly takes its produced census tract level outage scenarios to support the compound hazard resilience analysis, proves the power system model useful enough to support this research as well as the main findings.

Nevertheless, we perform further analysis for Hurricane Ike to test the assumptions on the distribution networks and protective deceives (Supplementary Fig. S9). To test the assumption on the distribution networks, we use the road network in Harris County to generate the MST structure of the power distribution networks and find that the specifics of the acyclic distribution network structure does not affect the accuracy of the global power system simulation (Fig. S9a, c). Although the distribution network topology is generally not available in a large domain, e.g., the whole east coast of the United States, the data is available for Harris County. The power system simulation results based on the true distribution networks are similarly accurate (Fig. S9d). Protective devices protect the distribution/transmission lines from overflow-induced physical damages; however, power outages caused by protective devices are usually restored in hours, which is not at the same scale as the post-hurricane power outage (weekly level). To further investigate how the protective devices could potentially impact the power system resilience, we assume all power lines are equipped with protective devices and incorporated the overload-induced component failures into the power outage modeling (the same as in ref. 75). We find that the recovery process is similar between the case considering cascading failure and the case not considering cascading failure (prevented by protective devices) (Fig. S9a, c). Also, although under the normal demand, there would be ~10% overloaded lines, when the demand is less than 95% of the normal demand (due to, e.g., hurricane evacuation), the overloading hardly happens (Fig. S9b). These findings illustrate that the impact of protective devices is relatively small on the daily-scale power outage and recovery under hurricanes, where a large number of physically damaged power system components often need days to weeks to be fully repaired. However, it is still worthy of note that the post-hurricane behavior of the electric power system, though simplified in this work to capture the census-level statistics of a power outage, is extremely complex, involving numerous physical processes on the transmission and distribution networks and a variety of devices and operations. Future research may develop more detailed models to better capture the physical mechanisms of post-hurricane power system failure.

**Network analysis and enhancement investigation.** After investigating the TC-blackout-heatwave compound hazard risk, we apply network analysis to investigate how the spatial pattern of a power outage is related to the network pattern of local power distribution sectors, using our large synthetic TC dataset. Specifically, we investigate the generalized scaling relationship between the probability of local failures and their impact on the global outage, proposed by ref. 8, to

understand the reliability of the power system. We analyze the connection between the power outage and local distribution network topology by linking the power outage rate to the mean length of local distribution sectors. These analyses support the aim of designing efficient hazard mitigation strategies.

Various strategies have been proposed to enhance the post-hurricane resilience of power networks[42], e.g., adding recovery resources, applying stricter structural criteria, and undergrounding network branches. Adding significant recovery recourses or applying stricter design criteria would raise daily costs while benefiting emergency recovery more than daily operation[40]. Thus, based on our findings, we design an undergrounding plan to enhance the resilience of the power system. Specifically, we propose greedily reducing the mean length of local distribution networks by protecting a small portion of wires close to root nodes of the distribution networks. We compare this strategy with undergrounding strategies that randomly bury transmission lines and distribution sectors (similar to the generally used uniform undergrounding strategies) to evaluate its efficiency in reducing future risk of TC-blackout-heatwave compound hazard. The results, based on the star-like distribution networks, is shown in Fig. 6. We test the effect of the assumption on the distribution networks by applying the analysis to the MST-based distribution networks and the true distribution networks and find similar results (Supplementary Fig. S10; e.g., for an enhancement rate of 5%, the expected percentage of Harris residents experiencing at least one longer-than-5-day TC-blackout-compound hazard is different only by 2% between MST and star-like based models and by 3% between the true distribution networks and star-like based networks).

## Data availability

The power network data (including topology data for transmission and distribution networks) and the historical power outage data were obtained from CenterPoint Energy, Inc. (www.centerpointenergy.com). The hurricane data were provided by Kerry Emanuel (MIT). The generated power system failure statistics are deposited to the NSF DesignSafe-CI under the ODC license (https://doi.org/10.17603/ds2-bqc0-sn69).

## Code availability

The code for simulating power system failures are deposited in the NSF DesignSafe-CI under ODC license (https://doi.org/10.17603/ds2-bqc0-sn69).

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

## Acknowledgements
K.F. and N.L. are supported by the U.S. National Science Foundation (1652448 and 2103754 as part of the Megalopolitan Coastal Transformation Hub) and C3.ai Digital Transformation Institute (C3.ai DTI Research Award). M.O. is supported by the National Natural Science Foundation of China (72074089, 51938004, and 71821001). We thank Tom Matthews (Loughborough University) for advising us on the heatwave analysis. We thank Kerry Emanuel (MIT) for providing the synthetic hurricane datasets. We also thank the reviewers for providing constructive suggestions.

## Author contributions
K.F. and N.L. designed the project and analyzed hurricane and climate data; K.F. and M.O. performed power outage and recovery analysis; K.F., N.L., and M.O. wrote the paper.

## Competing interests
The authors declare no competing interests.
