## [Peer Review File · Nature Communications]

Tropical cyclone-blackout-heatwave compound hazard resilience in a changing climateReviewers' Comments:

Reviewer #1:

Remarks to the Author:

Review of Hurricane-blackout-heatwave Compound Hazard Risk and Resilience in a Changing Climate by Kairui Feng et al.

Summary

This is a very interesting, timely, and ambitious study. It brings state of the art analysis from multiple fields to bear on a pressing climate risk. The paper is generally well presented and reaches conclusions that are both clearly explained and may be of immediate practical significance. It was enjoyable to read and I commend the authors on the tremendous effort that this paper must represent

However, there is a potential "catch" to the above. Arguably, there are too many links in the ambitious chain of analyses; the paper may be attempting to do too much – depending on how it is framed. Inappropriate assumptions at one stage threaten to undermine the entire study *if* the emphasis is on the sum of the individual parts, rather than the insights gleaned along the way. For example, as I highlight in Major Comments, I have concerns that the power system modelling (which I concede that I am not an expert in) may have unreasonable assumptions in the context of a changing climate (i.e. conditions beyond those that the model was calibrated for), and it is unclear how the insights gleaned here will be of benefit more widely (in particular for the global south – where hazard and vulnerability are higher – and in regions where surges/flooding are a major threat). To avoid this pitfall of overstating the conclusions (i.e. dwelling too heavily on the numbers at the end of the chain of processing), I suggest that, overall, more attention is given in the text to the insights along the way (e.g. compounding of heatwaves and TCs – the "joint" risk growing faster than the "marginal" risk). Combined with more clear caveating around assumptions in the methods (e.g. Major Comment [3]) and situating the case study's insights more within a global context, this would significantly improve the contribution. If the authors accept this suggestion, I recommend in particular that they revisit the abstract and elsewhere to ensure that the numbers/scenarios they provide are highlighted as very much illustrative/conditional upon their assumptions.

To be clear, overall I support publication of this work -- if the overall concern raised above, and the specific issues highlighted below. can be addressed or dismissed.

Major Comments

[1] Please explain early on why a detailed case study for Harris County is warranted. Relevant themes could include the disproportionately large increase in humid heat expected in sub-tropical locations, and perhaps the availability of datasets/models. Other references are of course available(!) but my own work will provide useful signposts with respect to the humid-heat dimension (perhaps as a starting point):

Matthews, T. Wilby, R.L. and Murphy, C., 2017. Communicating the deadly consequences of global warming for human heat stress. *Proceedings of the National Academy of Sciences*, 114(15), pp.3861-3866.

Matthews, T., 2018. Humid heat and climate change. *Progress in Physical Geography: Earth and Environment*, 42(3), pp.391-405.

[2] Methods: As noted in my annotations, I have some concerns/queries about the TC-heat calculations:

(a) How do you deal with the humidity when bias correcting the temperature? If you leave specific humidity unchanged, then the relative humidity would drop. Research elsewhere has shown that

biases tend to compensate, so models that underestimate temp may overestimate specific humidity (see Fischer and Knutti, 2013). If you bias correct only temperature, you may introduce more bias to the HI. A widely-used fix is to bias correct the HI, not the temp (see Matthews et al., 2017 -- cited above for an example of that).

(b) If I understand correctly, you correct GCM TC distributions based on the NCEP stochastic ensemble. How does the NCEP stochastic ensemble compare to reality? Perhaps this is explained in the relevant references, but please clarify.

[3] Power system modelling: I am concerned that there is an assumption of stationarity in here regarding the recovery response. I think it is reasonable to assume that the magnitude of the TC (through the damage to transport infrastructure it brings); and the severity of the heatwave (through impacting response workers' ability to discharge their duties) will influence the speed with which repairs can be made. In other words, the response in the power system modelling is (I think) currently treated as independent of the hazard, when in reality it will be coupled. The sample of two events for evaluation is interesting, but not sufficient to allay concern. At the very least this assumption (of stationarity) needs to be made clear somewhere

On a related note, and I think *partially* covered (in relation to repairing "wetted" power system components): is it really sensible to bury critical infrastructure when flooding is, in general, one of the most common consequences of TC landfall? I think this point is worth exploring in more detail when thinking about how the results from this case study apply more broadly. I suspect that there is other work in this space (TCs and resilient infrastructure) that deals with the trade-off between burial (protection from wind but exposed to flood), vs. elevated infrastructure?

Other references cited:

Fischer, E.M. and Knutti, R., 2013. Robust projections of combined humidity and temperature extremes. *Nature Climate Change*, 3(2), pp.126-130.

Please see the attachment for my minor comments (provided as annotation)

Signed

Tom Matthews

Reviewer #2:

Remarks to the Author:

This paper consists of two aspects. The first is on developing climate models and simulation on both hurricanes and heatwave aftermath. The second is to use the climate models to assess risks on power failures at a US city. The first part appears to be novel, although it may be helpful to motivate more why consider such a combination.

The second part of the paper generates scenarios on power failures based on Ref. 29. Several analysis is then conducted using the synthetic power failures. This includes results on failure scaling relating to customers, and grid enhancement through greedy grounding of power lines. This part is not well grounded technically. The specifics are provided below.

1. Simulating failures at a power grid is complex. The model used based on ref. 29 misses several key aspects. First, it is well known that failures trigger a large number of protective devices in the grid, which also results in interruptions of power supplies to customers. This is not considered in the simulation. Second, the distribution system only has accurate information from the transmission grid to the level of distribution substations. The rest of the distribution grid seems to be drawn from roads and residence; and affected customers are estimated through census data. As such, how customers are

affected by the failures can be quite inaccurate.

2. Because of the inadequacies discussed above, some of the results are not well grounded. For example, Fig. 5(b) on the impact of people is not well supported by the failure scenarios and assumptions on restoration. This is because the relationships between the failures, how long they last, and affected customers are not well established. Similarly, Fig. (6) is not well supported by the power grid simulation either, since the simulation cannot relate accurately the percentage customers with failures of long durations close to the root nodes. Fig. 4 and Fig 5(a) also draw relationships between the affected customers, the grid and restoration. The authors may want to examine those as well.

Overall, the paper is well written. The efforts made by the authors are evident. Unfortunately, the second part of the paper is technical flawed.

Reviewer #3:

Remarks to the Author:

This manuscript described the probability of hurricane-heatwave compound events, its impact on the power outages and blackouts, and paths to increasing resilience of the power grid and mitigate the risk of compound hazard. They focus on Harris County. There are three primary modeling components used in this study: synthetic hurricane model, heat wave index (HI), and the physics-based power grid model. This study uses a novel approach which forces a power system modeling with synthetic hurricanes (generated by climate informed hurricane models) and the associated heatwave conditions. As mentioned in the end of manuscript, this study demonstrates an interdisciplinary work for developing risk mitigation strategies to achieve resilient and sustainable communities. While I enjoyed the article, I have a few questions.

First of all, I wonder whether the modeling system used here can capture the "interdependence between hurricanes and heatwaves." From my understanding of this type of hurricane models (as well as reading the methodology section), the hurricane model uses monthly data (interpolated to daily resolution). While there are some stochastic processes that allow the model to capture the statistics of noises at higher (than monthly) temporal frequency, the output date (which day of the month) is not actually meaningful. On the other hand, the heat wave is defined by the daily data (L417). Can you further explain the physical connections that result in more hurricane-heatwave events at Harris City? (or this is simply due to that there will be more high impact hurricanes and more heatwaves, so the probability of them happening together increases).

My second question is what is the impact of the bias-correction to the resulting climate change impact? Assuming that the observed Houston is 2 storm/year and your model is 1.2 storm/year for historical period and 3/year for rcp period with 1.8 storm/yr increase. Now to adjust the model biases, did you keep the 1.8 storm/year increase or increased by the ratio between 1.2 and 2 storm/year. In other words, how the climate change difference changes with the bias-correction, and what is the impact to the downstream analysis.

Lastly, is the wind gust considered to force the power system model? From L442, I think you are using the mean maximum wind (1 minutes or 10 minutes wind). Is there a way to add gust estimation here? (I am somehow under an impression that we should use wind gust instead of mean wind for calculating hurricane induced power outage.)

One minor comment: L20 please make it clear that you meant the hurricane-heatwave events, not just heatwaves.

**Hurricane-blackout-heatwave
compound hazard risk and resilience in a changing climate**
Kairui Feng, Ouyang Min, Ning Lin

Response to the reviewers

Reviewer #1 (Remarks to the Author):

Review of Hurricane-blackout-heatwave Compound Hazard Risk and Resilience in a Changing Climate by Kairui Feng et al.

Summary

This is a very interesting, timely, and ambitious study. It brings state of the art analysis from multiple fields to bear on a pressing climate risk. The paper is generally well presented and reaches conclusions that are both clearly explained and may be of immediate practical significance. It was enjoyable to read and I commend the authors on the tremendous effort that this paper must represent

However, there is a potential “catch” to the above. Arguably, there are too many links in the ambitious chain of analyses; the paper may be attempting to do too much – depending on how it is framed. Inappropriate assumptions at one stage threaten to undermine the entire study *if* the emphasis is on the sum of the individual parts, rather than the insights gleaned along the way. For example, as I highlight in Major Comments, I have concerns that the power system modelling (which I concede that I am not an expert in) may have unreasonable assumptions in the context of a changing climate (i.e. conditions beyond those that the model was calibrated for), and it is unclear how the insights gleaned here will be of benefit more widely (in particular for the global south – where hazard and vulnerability are higher – and in regions where surges/flooding are a major threat). To avoid this pitfall of overstating the conclusions (i.e. dwelling too heavily on the numbers at the end of the chain of processing), I suggest that, overall, more attention is given in the text to the insights along the way (e.g. compounding of heatwaves and TCs – the “joint” risk growing faster than the “marginal” risk). Combined with more clear caveating around assumptions in the methods (e.g. Major Comment [3]) and situating the case study’s insights more within a global context, this would significantly improve the contribution. If the authors accept this suggestion, I recommend in particular that they revisit the abstract and elsewhere to ensure that the numbers/scenarios they provide are highlighted as very much illustrative/conditional upon their assumptions.

Response: Thanks to the reviewer for the encouragement. We agree that the study is ambitious in integrating a number of parts and accounting for various factors. We have addressed the specific uncertainties the reviewers pointed out and added significantly to Discussion and Supplementary (please see below). Limitations are also further addressed in Methods (see below). And we take the reviewer’s suggestion to largely rewrite the abstract to highlight qualitative conclusions (e.g., adding intermediate results, change factors in addition to specific changes, and perspectives on resilience and joint risk) and uncertainties (e.g., power grid aging, impeded power recovery under heatwaves and extreme weather, and rapid coastal development). The broader application and global context are added to the end of Discussion. We believe the main contribution of the study is now better presented.

To be clear, overall I support publication of this work -- if the overall concern raised above, and the specific issues highlighted below, can be addressed or dismissed.

Major Comments

[1] Please explain early on why a detailed case study for Harris County is warranted. Relevant themes could include the disproportionately large increase in humid heat expected in sub-tropical locations, and perhaps the availability of datasets/models. Other references are of course available(!) but my own work will provide useful signposts with respect to the humid-heat dimension (perhaps as a starting point):

Matthews, T. Wilby, R.L. and Murphy, C., 2017. Communicating the deadly consequences of global warming for human heat stress. *Proceedings of the National Academy of Sciences*, 114(15), pp.3861-3866.

Matthews, T., 2018. Humid heat and climate change. *Progress in Physical Geography: Earth and Environment*, 42(3), pp.391-405.

Response: Thanks for the valuable comments. We have added the point and reference Matthews et al. 2017 to the Introduction "Harris County has the highest population density along the Gulf Coast and, as located in the subtropics, may face disproportionately large increase in heatwaves [30] and TCs [31] in a warming climate." and Discussion "The analysis results for Harris County obtained in this study may be qualitatively representative for coastal megacities in the subtropical regions of the U.S., but the developed methodological framework can be applied to quantify power system resilience and develop risk mitigation strategies for other TC-prone areas".

[2] Methods: As noted in my annotations, I have some concerns/queries about the TC-heat calculations:

(a) How do you deal with the humidity when bias correcting the temperature? If you leave specific humidity unchanged, then the relative humidity would drop. Research elsewhere has shown that biases tend to compensate, so models that underestimate temp may overestimate specific humidity (see Fischer and Knutti, 2013). If you bias correct only temperature, you may introduce more bias to the HI. A widely-used fix is to bias correct the HI, not the temp (see Matthews et al., 2017 -- cited above for an example of that).

Response: We originally bias corrected only the temperature, considering it is the most important component in HI. Thanks very much for pointing us to this important issue. We have now changed to bias correct HI following Matthews et al. 2017. The method, references (Matthews et al. 2017 and Fisher and Knutti 2013), and results are updated. The compound hazard estimations we obtain by bias-correcting HI now are slightly larger than the original result, e.g., the expected percent of residents who may experience at least one longer-than-5-day heatwave without power in future climate would be 18.2% instead of the original 15.1%. But the conclusions are all qualitatively the same.

(b) If I understand correctly, you correct GCM TC distributions based on the NCEP

stochastic ensemble. How does the NCEP stochastic ensemble compare to reality? Perhaps this is explained in the relevant references, but please clarify.

Response: We use the TC data simulated by Emanuel 2017 for the study region. Emanuel 2017 used the statistical-deterministic model developed by Emanuel et al. 2006, 2008. The model was evaluated in Emanuel et al. 2006 (and elsewhere) using NCEP simulation compared with observation. We have added to the Method “The model, when driven by reanalysis environment, is shown to generate synthetic storms that are in statistical agreement with observations [56], and it has been widely used to study TC wind, storm surge, and rainfall hazards under climate change effects (e.g., 33,36).”

Emanuel et al. (2006). A Statistical Deterministic Approach to Hurricane Risk Assessment. *Bulletin of the American Meteorological Society*. 87. 299-314. 10.1175/BAMS-87-3-299.

[3] Power system modelling: I am concerned that there is an assumption of stationarity in here regarding the recovery response. I think it is reasonable to assume that the magnitude of the TC (through the damage to transport infrastructure it brings); and the severity of the heatwave (through impacting response workers’ ability to discharge their duties) will influence the speed with which repairs can be made. In other words, the response in the power system modelling is (I think) currently treated as independent of the hazard, when in reality it will be coupled. The sample of two events for evaluation is interesting, but not sufficient to allay concern. At the very least this assumption (of stationarity) needs to be made clear somewhere

Response: Thanks for the valuable comments. Indeed, the recovery operation may change in the future, and such changes can be incorporated in our physics-based modeling framework. As future operation policy and/or technology may change in less predictive ways, we did not include this factor in the main analysis presented in the Results. But we have now added a sensitivity analysis in Discussion: “The power system operation may also be affected by extreme heat. According to the Occupational Safety and Health Administration (OSHA) criteria, outdoor workers can work for only limited hours (<75% normal hours) under extreme heat and humidity (HI>39.4°C). Assuming there is no advanced technology to improve outdoor working condition and the recovery operation follow the OSHA criteria, the expected percent of Harris County residents experiencing at least one longer-than-5-day TC-blackout-heatwave compound hazard in a 20-year period under the future climate could increase from 18.2% (workers working normally) to 23.3% (see Supplementary Fig. S7).”

On a related note, and I think *partially* covered (in relation to repairing “wetted” power system components): is it really sensible to bury critical infrastructure when flooding is, in general, one of the most common consequences of TC landfall? I think this point is worth exploring in more detail when thinking about how the results from this case study apply more broadly. I suspect that there is other work in this space (TCs and resilient infrastructure) that deals with the trade-off between burial (protection from wind but exposed to flood), vs. elevated infrastructure?

Response: The burying of power system has its engineering criteria. Usually, the power lines will be protected by anti-water pipelines (Ref. 35). We have now added this to Introduction as “burying parts of the power network covered by anti-water pipelines [39-41], see Methods”. The following figure shows the pipelines that used to protect the power lines (Fig.

1R). Also, we collected and investigated several post-TC news and reports for Coastal cities and found that flooding-induced failures of underground wires are seldomly reported.

Fig. R1. The underground pipelines used to cover the electric lines (from IEW Constructing Group <https://www.iewconstructiongroup.com/project/electrical-underground/>)

Other references cited:

Fischer, E.M. and Knutti, R., 2013. Robust projections of combined humidity and temperature extremes. *Nature Climate Change*, 3(2), pp.126-130.

Please see the attachment for my minor comments (provided as annotation)

Response: We thank the reviewer for the very detailed comments. We have addressed each annotated point and we believe the manuscript is much improved.

Signed

Tom Matthews

Reviewer #2 (Remarks to the Author):

This paper consists of two aspects. The first is on developing climate models and simulation on both hurricanes and heatwave aftermath. The second is to use the climate models to assess risks on power failures at a US city. The first part appears to be novel, although it may be helpful to motivate more why consider such a combination.

Response: Thanks the reviewer for the comment. Indeed, the main focus of this study is to combine climate modeling and power system modeling to address the TC-blackout-heatwave compound risk. As the climate gets warmer, both heatwaves and TCs may get stronger. Matthews et al. (2019; published in *Nature Climate Change*) first studied the potential compound hazard of heatwaves and TCs, but they didn't study the compound risk. Here we combine the compound hazard projections (including also TC projections that were not

included in Matthews et al.) and power failure and recovery modeling to study the compound risk and resilience. We have revised the abstract, introduction, and elsewhere to make this motivation and goal clearer.

The second part of the paper generates scenarios on power failures based on Ref. 29. Several analysis is then conducted using the synthetic power failures. This includes results on failure scaling relating to customers, and grid enhancement through greedy grounding of power lines. This part is not well grounded technically. The specifics are provided below.

Response: We thank the reviewer for challenging our model on many fronts. To address the reviewer's concerns, we have made great efforts (over the past six months) to check various model assumptions through numerous additional experiments, including adding validation at the census tract level, performing sensitivity analysis on protective devices and cascading failures, and checking the assumption on distribution networks (please see below for details). Although our original conclusions still hold, we believe these additional analysis and literature review have better addressed the modeling scope and limitations.

1. Simulating failures at a power grid is complex. The model used based on ref. 29 misses several key aspects.

Response: We agree that simulating power failure and recovery is complex, and the model we use, although physics-based, cannot cover all the details. However, the adopted model may be the best model that we can use to simulate the power outage physically (e.g., capture the vulnerability of power poles under different wind speed rather than directly scaling the damage) under thousands of TC cases. We have added to the Method: "The power system model we use here, though physics-based, cannot resolve all details. Given that distribution network and protective device data are generally not available, star-like network [34] and minimal spanning tree [71] models are usually used to generate the distribution networks and features that may have minor effects under TC conditions including protective devices are usually neglected [72]. Based on these assumptions, the adopted simulation framework is arguably the best model that we can use to capture the power outage and recovery process at a meso-scale level, e.g., for each zip code or census tract, rather than at the individual household level, for risk and resilience analysis. The model using star-like distribution networks and neglecting protective devices is validated for Hurricane Ike and Harvey at both the county level (Fig. 2) and census tract level (Fig. S2)."

First, it is well known that failures trigger a large number of protective devices in the grid, which also results in interruptions of power supplies to customers. This is not considered in the simulation.

Response: Thanks for this valuable comment. Protective devices will be triggered when power line sector is broken or over load happens. The triggering of protective devices is vital in daily power operations. Many utility companies use protective device data as a proxy to individual level power outage data, as in Ji et al. 2016 (Ref. 8; published in Nature Energy). Given data on that detailed level, Ji et al. made a significant contribution to the field for understanding post-TC power outage, including revealing the scaling law of local level power outage. The analysis in Ji et al. shows on the event level, few significant power outage event dominants major power outage. This stimulated our research on studying retrofitting power systems.

As protective device data is generally not available, we didn't include it in the methodology, but we have now performed experiments to test the impact. We have added to the Method: "To investigate how the protective devices could potentially impact the power system resilience, we assume all power lines are equipped with protective devices and incorporated the overload-induced component failures into the power outage modeling (the same as in Ref. [72]). We find that the recovery process is similar between the case considering cascading failure and the case not considering cascading failure (prevented by protective devices) (Fig. S9a). Also, although under the normal demand there would be ~10% overloaded lines post the hurricane, when the demand is less than 95% of the normal demand (e.g., due to hurricane evacuation) the overloading hardly happens (Fig. S9b). Thus, the impact of protective devices is relatively small in post-hurricane power system resilience."

In our paper, the power response model against component failures was adapted from an earlier widely cited article by Hines et al. (2010) (Ref. 72). This article introduced a cascading failure model to approximate the process that automate controls (each link has a relay that removes it from service if its current exceeds a limit) and operators to balance supply and demand during extreme events. Many other papers follow the same simulation principle for power system reliability analysis, e.g., Yang 2017 a (published in Physics Review Letters, Ref. 68) and Yang 2017 b (published in Science, Ref. 69).

The description of the model proposed by Hines is copied as following for explanation: "After a component fails the model recalculates the power flow and advances to the time at which the next component fails or quits if no further components are overloaded. If a component failure separates the grid into unconnected sub-grids, the following process is used to re-balance supply and demand. If the imbalance is small, such that generators can adjust their output by not more than 10% and arrive at a new supply/demand balance, this balance is achieved through generator set-point adjustments. If this adjustment is insufficient, the smallest generator in the sub-grid is shut down until there is an excess of load. If there is excess load after these generator adjustments, the simulator curtails enough load to balance supply and demand".

The difference between the Hines' model and the adapted model lies in the treatment of overload-induced component failures in each sub-grid after the initial failures. To test the impact, we have taken this into account (with more computational cost) and updated the power response model (to be the same as the Hines' model), and the results are almost unchanged. We further investigated how much this additional consideration of "overload-induced component failures" affects the estimation of the number of customers with power. As shown in Fig. R2 (added to the Supplementary), the difference is extremely small. We also note that before landfalls of TCs, residents are usually informed in advance to stay at home or evacuated, and then the power demand runs far lower than its limit state. If we reduce the power demand by 25%, the number of overload components under different TC scenarios becomes almost zero, as shown in Fig. R3 (added to the Supplementary).

Figure.R2. Post-Ike restoration curves averaged over 100 runs in the case of considering cascading failure and not considering cascading failure when the distribution networks are modelled star-like (a) and MST-based (b).(x axis reads as hours)

Figure.R3. Fraction of overloaded lines during the restoration process when the power operation model considers the cascading failure. When the power system demand decreases to 70% of its normal demand, cascading failures will not occur during the whole restoration process. (x axis reads as hours). Discontinuous lines are because cascading failures will not occur at some time.

We also note that the most realistic models that capture the post-attack behaviors often involve the dynamically-induced transients and/or the nonlinearity of alternative current (AC) power flow, such as the AC power blackout model developed at the University of Manchester, representing a range of interactions and including cascade and sympathetic tripping of transmission lines, heuristic representation of generator instability, under-frequency load shedding, post-contingency re-dispatch of active and reactive resources, and emergency load shedding to prevent a complete system blackout caused by a voltage collapse (Rios et al., 2002; Kirschen, et al., 2004). But those models are more complicated and comprehensive than the direct current (DC) flow model. Without parallelization and graph theory, the original MatPower takes over 5 mins to do one simulation for post-TC recovery modelling for Harris County. Our modified DC model takes only 2 seconds. To the best of our knowledge, this is the only model that is able to massively simulate (there are over 40,000 simulations done in this paper) the failure of every single power sector while giving good census tract level

validation result (not losing a lot of power system information).

C Ji, et al., Large-scale data analysis of power grid resilience across multiple us service regions. *Nat. Energy* 1, 16052 (2016)

Yang, Y., Nishikawa, T., Motter, A.E., 2017a. Vulnerability and cosusceptibility determine the size of network cascades. *Phys. Rev. Lett.* 118 (4) 048301.

Yang, Y., Nishikawa, T., Motter, A.E., 2017b. Small vulnerable sets determine large network cascades in power grids. *Science* 358 (6365).

Hines, Paul, Eduardo Cotilla-Sanchez, and Seth Blumsack. "Do topological models provide good information about electricity infrastructure vulnerability?." *Chaos: An Interdisciplinary Journal of Nonlinear Science* 20.39 (2010): 033122.

Rios MA, Kirschen DS, Jayaweera D, Nedic DP, Allan RN. Value of security: modeling time-dependent phenomena and weather conditions. *IEEE Trans Power Syst* 2002;17(3):543–8.

Kirschen DS, Jayaweera D, Nedic DP, Allan RN. A probabilistic indicator of system stress. *IEEE Trans Power Syst* 2004;19(3): 1650–7.

Second, the distribution system only has accurate information from the transmission grid to the level of distribution substations. The rest of the distribution grid seems to be drawn from roads and residence; and affected customers are estimated through census data. As such, how customers are affected by the failures can be quite inaccurate.

Response: Thanks for raising the important issue. We do not employ detailed level of power system down to residence level. Our aim is to simulate the power failure and resilience processes at the census tract level. We focus on statistical description of census tract level power outage pattern. We have now made this scope clearer (adding “census tract” at appropriate places in Introduction, Results and Method). We believe such an approach is appropriate for climate risk research supporting decision making. Data at the residential level is generally not available, and simulations at that detail is not feasible for climate risk analysis here involving ~40,000 simulations. (The affected customers are not estimated through census data, but released by CenterPoint Energy.)

As detailed distribution system data is often not available, as an alternative, we used star-like structure to approximate the distribution network, and each intersection along the secondary road segments is extracted as a proxy distribution node connected to the nearest transmission substation. Despite it may deviate from the reality, assigning appropriate values to the two adjusted parameters (α, β) of the model can well reproduce the historical outage and recovery process for the real system under Hurricanes Harvey and Ike. In the original manuscript, we only provided the power outage validation for Harris County. In the revised manuscript, we further provided the validation results for all the census tracts in Harris County (Fig. R4, added to the Supplementary). The x-axis shows the observation level of initial power outage while the y-axis shows the simulated initial power outage level. Different points show different census tracts. We have added to the Results: “The model results at the census tract level for Hurricanes Ike and Harvey compare also relatively well with observations (Supplementary Fig. S2), with a mean error over all census tracts less than 10%.”

Fig. R4. The initial power outage rate after Hurricane Ike and Harvey under simulation comparing with observation on census tract level

To further check the star-like distribution assumption, we have used the local road network to construct the minimum spanning tree (MST) to connect distribution nodes and the transmission substation (Fig. S1), and took this MST with the shortest length as a local distribution network (as shown in Fig. R5, added to Supplementary). Using the historical outage and recovery data, we calibrated the new value settings for α , β ($\alpha=0.9$, $\beta=1$), with the mean error over all census tracts less than 10%, and the simulated restoration curve also fits the real curve well. Thus, using the MST-based distribution networks results in similar results (as shown in Fig. R6, incorporated in Supplementary). We have added to the Method: “To test the assumption on the distribution network, we use Harris road network to generate the minimum spanning tree based structure of the power distribution network (Fig. S9c) and find that the specifics of the acyclic distribution network structure does not affect the accuracy of the global power system simulation (Fig. S9d).”

Figure. R5. (a) A snapshot of the real distribution network in Netherlands. The distribution lines are deployed along local urban streets. (b) an example of the MST-based distribution network originated from a load substation in Harris power transmission system; The local road streets (including all classes of road segments) are also displayed in (b).

Fig. R6. MST-based distribution network, and the calibrated parameters are updated as $\alpha=0.9$, $\beta=1$.

Moreover, note that the real distribution system falls in between the star structure and MST structure in terms of the length. As long as the system is close to an acyclic structure, the findings will not be affected. In Supplementary, we proved that for all the acyclic local distribution network structures, the expected power outage under random attack would be proportional to the mean harmonic length of the distribution network branches. This could be used to support the simplification on local distribution network data.

Finally, it is possible the detailed distribution networks may affect the results at the household level. However, again the applied model aims to simulate the power failure and resilience processes on the census tract level.

2. Because of the inadequacies discussed above, some of the results are not well grounded. For example, Fig. 5(b) on the impact of people is not well supported by the failure scenarios and assumptions on restoration. This is because the relationships between the failures, how long they last, and affected customers are not well established.

Response: As stated in the above response, the adopted model itself has been validated using the real outage and recovery data at the census tract level, and is good enough to simulate the power failure and resilience processes well at the census tract level in Fig. 5b. More validation results are shown in Fig. R3. Fig. 5(b) is used to uncover the trend between the affected customers at the census tract level and the mean length of the distribution network sectors, and the uncertainty is mainly originated from the repair priority rules and also the spatial heterogeneous distributions of gust wind speed and its impact on power system components.

Similarly, Fig. (6) is not well supported by the power grid simulation either, since the simulation cannot relate accurately the percentage customers with failures of long durations close to the root nodes.

Response: Without precise data on the distribution network, we agree that it is impossible to accurately capture the relation between the affected customers and their locations. However, in Supplementary, we proved that for all the acyclic local distribution network structures, the expected power outage under random attack would be proportional to the mean harmonic length of the distribution network branches. The enhancement of distribution network on root nodes would directly contribute to the mean harmonic length of the distribution network branches for each substation. This also justifies the trend in Fig. (6). Also, besides the star-like structure, we further made additional experiments for the MST-based structure of the power distribution network, which gives similar results as shown in Fig. R7 (Added to Supplementary). These results show the retrofitting efforts (the percentage of distribution network undergrounded) affecting the 5-day compound hazard level under the future climate. These two different distribution network generators give similar retrofitting effects under the same retrofitting strategy. We have added to the Method: “The results, based on the star-like distribution network, is shown in Fig. 6. We test the effect of the assumption on the distribution network by applying the analysis to the MST based distribution network and find similar results (Supplementary Fig. S10; e.g., for an enhancement rate of 5%, the expected percent of residents experiencing at least one longer-than-5-day TC-blackout-compound hazard is different only by 2% between MST and star-like based models).”

Fig.R7. The effect of undergrounding under different distribution network assumptions

Fig. 4 and Fig 5(a) also draw relationships between the affected customers, the grid and restoration. The authors may want to examine those as well.

Response: As discussed in the above, the power outage and recovery model employed in this paper is validated with the real affected customers at the census tract level during the whole recovery process. This data was released by CenterPoint Company. More validation results are shown in Fig. R4. After the validation, results are aggregated and displayed at the census-tract level. Scaling law results in Fig. 5a compares well with those in Ji et al. for Hurricane Sandy using household level data, as discussed in Results.

Overall, the paper is well written. The efforts made by the authors are evident. Unfortunately, the second part of the paper is technical flawed.

Response: We thank the reviewer for the critical comments. The authors have carefully re-checked the model and conducted numerous additional experiments to address the concerns from the reviewer. Results point out that the employed model is good enough to simulate the power outage and recovery at the census tract level. Related explanations have been added to the revised manuscript to enhance clarity. Thanks for helping us improve the manuscript.

Reviewer #3 (Remarks to the Author):

This manuscript described the probability of hurricane-heatwave compound events, its impact on the power outages and blackouts, and paths to increasing resilience of the power grid and mitigate the risk of compound hazard. They focus on Harris County. There are three primary modeling components used in this study: synthetic hurricane model, heat wave index (HI), and the physics-based power grid model. This study uses a novel approach which forces a power system modeling with synthetic hurricanes (generated by climate informed hurricane models) and the associated heatwave conditions. As mentioned in the end of manuscript, this study demonstrates an interdisciplinary work for developing risk mitigation strategies to achieve resilient and sustainable communities. While I enjoyed the article, I have a few questions.

First of all, I wonder whether the modeling system used here can capture the

“interdependence between hurricanes and heatwaves.” From my understanding of this type of hurricane models (as well as reading the methodology section), the hurricane model uses monthly data (interpolated to daily resolution). While there are some stochastic processes that allow the model to capture the statistics of noises at higher (than monthly) temporal frequency, the output date (which day of the month) is not actually meaningful. On the other hand, the heat wave is defined by the daily data (L417). Can you further explain the physical connections that result in more hurricane-heatwave events at Harris City? (or this is simply due to that there will be more high impact hurricanes and more heatwaves, so the probability of them happening together increases).

Response: Thanks for the comment. This is a very important question. First, most climate variables for TC projection vary relatively slowly in time (e.g., MIP, ocean conditions) and thus monthly data is used to drive the TC model. However, some other variables, especially the environmental wind, which affects storm tracks and intensity through wind shear effect, varies much more quickly, and thus daily wind is used and stochastic (higher frequency) winds are generated based on the daily statistics to drive the TC model. (This wind model is derived in the model reference Ref. 32.) Thus, temporal variation at the daily scale is to some extent captured (the date of the month does carry climatology), but with uncertainties, and thus large numbers of simulations are used to account for such uncertainties and probabilistic estimation of risk is pursued downstream. We combined the storm and heatwave datasets through matching their dates, and thus the results of more TC-heatwave events is due to both higher impact TCs and heatwaves and their joint climatology. As a result, the joint risk is larger than the marginal risk. We have revised the sentence on this point to “The 23-time increase of TC-blackout-heatwave compound hazard, which is larger than simply folding the two factors together as if TCs and heatwaves are climatologically uncorrelated (21 times), indicates that large TCs and long-lasting heatwaves will be more likely to co-occur under climate change.” This point is also now added to the abstract.

Second, TCs and heatwaves may interact with each other physically (in addition to the joint climatology change under climate change). As there is no study yet on their physical interaction, we account for this effect empirically. As mentioned in the Method: “we add the composite impact of TC passage to the meteorological variables used to calculate the HI, where the composite impact is estimated based on historical data (Fig. 3a in ref. [27]).” As the data shows the HI drops after TC passage, we find (in Method): “Accounting for this correction reduces the HI values within 5 days of TC passage and thus the probability of 5-day compound TC-blackout-heatwave events defined in this study.” Thus, accounting for this empirical interdependence reduced the risk compared to that accounting for only the joint climatology as discussed above.

K Emanuel, R Sundararajan, J Williams, Hurricanes and global warming: Results from downscaling IPCC AR4 simulations. *Bull. Am. Meteorol. Soc.* 89, 347–368 (2008).

T Matthews, RL Wilby, C Murphy, An emerging tropical cyclone–deadly heat compound hazard. *Nat. Clim. Chang.* 9, 602–606 (2019)

My second question is what is the impact of the bias-correction to the resulting climate change impact? Assuming that the observed Houston is 2 storm/year and your model is 1.2 storm/year for historical period and 3/year for rcp period with 1.8 storm/yr increase. Now to adjust the model biases, did you keep the 1.8 storm/year increase or increased by the ratio

between 1.2 and 2 storm/year. In other words, how the climate change difference changes with the bias-correction, and what is the impact to the downstream analysis.

Response: We use the ratio, as mentioned in Method: “Specifically, for each GCM-driven projection, we bias-correct the projected storm frequency for the future climate by multiplying it by the ratio of the NCEP estimated frequency (where the NCEP frequency was calibrated to be 1.5 times/year for Houston using historical data [57]) and GCM estimated frequency for the historical climate, assuming no change in the model bias over the projection period, following refs. [57,58].” The assumption is used in previous studies. The ratio (rather than the difference) is used considering the regression of Poisson model (of storm arrives assuming storms are conditionally independent) is a log-linearized model. Therefore, the systematic bias is added to a logged factor and contribute to the frequency in a multiplicative way. The storm frequency does have significant impact on the downstream analysis. The projection of TC frequency is quite uncertain, with most models project a decrease or no change and a few models, including the one use here, predicting an increase. To test the sensitivity, we performed all analysis assuming no change in the storm frequency and find a reduced risk from 18.2% (assuming projected increase of storm frequency) to 11.2%. We have this included in the Discussion: “Even if we account for the uncertainty in the prediction of storm frequency [54] and remove the predicted increase in the storm frequency for the study area (by applying the frequency in the historical climate), the impact percentage would still increase significantly, to 11.2% towards the end of the 21st century (Supplementary Fig. S5).”

N Lin, RE Kopp, BP Horton, JP Donnelly, Hurricane Sandy’s flood frequency increasing from year 1800 to 2100. *Proc. Natl. Acad. Sci.* 113, 12071–12075 (2016).

Lastly, is the wind gust considered to force the power system model? From L442, I think you are using the mean maximum wind (1 minutes or 10 minutes wind). Is there a way to add gust estimation here? (I am somehow under an impression that we should use wind gust instead of mean wind for calculating hurricane induced power outage.)

Response: Thanks. Good catch. We did use the wind gust but forgot to explain it. We have now added to the Method when describing the hurricane wind field modeling “...and converting mean winds to wind gusts using gust factors [63],...”, and to the Method when describing the power system modeling “Given the TC wind hazard (i.e., local maximum wind gust during the storm; ... the power grid failure model first applies...”

One minor comment: L20 please make it clear that you meant the hurricane-heatwave events, not just heatwaves.

Response: Thanks. Yes, it is hurricane-heatwave events. More specifically, we have changed it to “TC-blackout-heatwave”.

Reviewers' Comments:

Reviewer #1:

Remarks to the Author:

The authors have addressed my major concerns, and I think the paper is now a valuable addition to our understanding in this field. I have included some minor comments in the attachment.

Reviewer #2:

Remarks to the Author:

The authors have made commentable efforts revising the manuscript. The first part of the work on the combined risks is now well motivated and clearly described.

The efforts made by the authors are appreciated on revising the power grid simulation. However, I am afraid to say that the second part is not technically well grounded.

- As pointed out in the previous comments, a major component missing in the simulation is the distribution grid. The available data to this work is on transmission substations and road networks (and impact on community)(Ref 34).

To make up for the information at the distribution grid, the revision draws information from prior publications to conduct simulation studies. However, these publications (Refs. 68, 69, 72) do not seem to provide key information on how failures occurred at the distribution grid. Refs 8 did but not on simulation.

- Specifically, the study on protective devices does not seem to be based on right scenarios. To protect the grid, power companies have installed all kind of protective devices, i.e., at the substations and transformers, not just those on the transmission lines. Therefore, it seems to be a big jump to claim that protective devices do not matter to failure patterns. Additionally, fault currents rather than overload often occur during weather events. Focusing on overload induced failures may be misleading for the distribution grid.

--It is intuitive to consider climate change at the county level. However, aggregating over census track does not seem to solve the problem on what we do not know about failures. What key factors to include in the simulation may require more knowledge about how failures occurred at the grid.

Reviewer #3:

Remarks to the Author:

The authors have addressed my previous comments which are mostly related how they simulated the compound hazards. I do not have further questions.

Hurricane-blackout-heatwave
Compound Hazard Risk and Resilience in a Changing Climate
Kairui Feng, Ouyang Min, Ning Lin

Response to the reviewers

Reviewer #1:

The authors have addressed my major concerns, and I think the paper is now a valuable addition to our understanding in this field. I have included some minor comments in the attachment.

Thanks to the reviewer for the positive comments. Thanks for checking on the details again. We've updated the main text based on the comments.

Reviewer #2:

The authors have made commendable efforts revising the manuscript. The first part of the work on the combined risks is now well motivated and clearly described.

Thanks to the reviewer for acknowledging the scientific contribution of our paper on climate impacts.

The efforts made by the authors are appreciated on revising the power grid simulation.

Thanks to the reviewer for appreciating our efforts.

However, I am afraid to say that the second part is not technically well grounded.

- As pointed out in the previous comments, a major component missing in the simulation is the distribution grid. The available data to this work is on transmission substations and road networks (and impact on community)(Ref 34).

To make up for the information at the distribution grid, the revision draws information from prior publications to conduct simulation studies. However, these publications (Refs. 68, 69, 72) do not seem to provide key information on how failures occurred at the distribution grid. Refs 8 did but not on simulation.

Thanks to the reviewer for the valuable comment. We fully agree that the post-hurricane outage and recovery modeling of the whole electric power system, including transmission system and distribution system, is very complicated, involving the analytical modelling of failure, recovery and cost with respect to time, geolocation, system location, customers, and their interdependencies (Ji et al. 2016). This complexity drives the emergence of statistical modeling, which utilizes various real data sources on power outage and recovery post-hurricanes and trains a spatiotemporal process model to predict power outage and recovery at various census levels. Physics-based models, which involves descriptions of network topologies and dynamics of the power system, have also been developed. Although providing more detailed information than the statistical models, current physics-based models still do not describe the details of the networks (i.e., distribution network) in a deterministic way, mainly due to the lack of data/information on the distribution network and numerous devices and other components. Thus, these models are used to estimate power failure statistics at the

aggregated level (i.e., census tract). Our analysis shows that the physics-based model can capture the environmental forces on power systems to match the observation, which is also often at the aggregated level (i.e., census tract). However, as the statistician George Box said, “all models are wrong, but some are useful”. The physics-based model employed here, although cannot reveal the failures deterministically at the distribution grid (again, mainly due to the fundamental limitation of distribution grid data), is sufficient for our purpose/scope of the investigation the impact of global climate change on the power grid reliability and resilience. The reviewer is right that Refs. 68, 69, 72, similar to our approach cannot predict failure deterministically at the distribution grid. To our knowledge, no simulation model has showed the capability of predicting failure deterministically at the distribution grid level, as the input data of the distribution grid is usually not available. And Ref. 8 does not provide a simulation model.

Currently there are no perfect distribution network data, so all the research may follow some distribution network generation methods and match the statistical results with macro-observations. The newest work to detailly synthetic distribution grid is done by Li et al. 2020 (led by NAE member Tomas Overbye). This is a large project sponsored by Department of Energy to develop synthetic power grid to be used as the real grid. However, they also don't have detailed distribution grid information rather than statistics and could only match their synthetic network metrics with real world metrics. Therefore, to our knowledge, there is not a way to run the post-hurricane power outage model on a real distribution network. The newest work to simulate power outage for hazard on power distribution systems is Zhai et al. 2021 (led by Prof. Guikema from Umich, who is also an expert in post-hurricane power outage). Both Li et al. 2020 and Zhai et al. 2021 generate synthetic power systems assuming MST and/or star networks. In our last response, we've shown that MST and star distribution networks give similar simulation results under hurricane scenarios (included in the main text and supplementary; also shown in Fig. R2 here). This could further convince us that the main approach in this paper could support its findings. We have added these references in Method as “The power system model we use here, though physics-based, cannot resolve all details. Given that distribution network and protective device data are generally not available, star-like network [35] and minimal spanning tree (MST) [72] models (and models combining these two [72, 73]) are usually used to generate synthetic distribution networks and features that may have minor effects under hurricane conditions including protective devices are usually neglected [72-74].”.

Figure.R2. Post-Ike restoration curves averaged over 100 runs in the case of considering cascading failure and not considering cascading failure when the distribution networks are modelled star-like (a) and MST-based (b).(x axis reads as hours)

Zhai, C., Chen, T. Y. J., White, A. G., & Guikema, S. D. (2021). Power outage prediction for natural hazards using synthetic power distribution systems. *Reliability Engineering & System Safety*, 208, 107348.

Li, H., Wert, J. L., Birchfield, A. B., Overbye, T. J., San Roman, T. G., Domingo, C. M., ... & Palmintier, B. (2020). Building highly detailed synthetic electric grid data sets for combined transmission and distribution systems. *IEEE Open Access Journal of Power and Energy*, 7, 478-488.

In summary, as (1) we study the census-tract-level outage scenarios several days long to combine with the heatwave model (also at census-tract-level) for estimating the risk of customers (at the census-tract level) experiencing heatwave after hurricane-induced power outage and (2) the adopted physical model was validated to predict census-tract-level outage and recovery well using real data, we believe our physics-based model is sufficient to support this research as well as the main findings.

Nevertheless, to respond to the reviewer's concern, we considered other possible approaches for estimating the hurricane-blackout-heatwave risk. Thus, we have developed a statistical model (as an alternative approach) to test our findings. We gathered census-tract-level power outage data and apply logistic regression of the total outage number in each census to tree types, maximum wind, maximum rainfall, land type and elevation for hurricane Ike and Harvey. We then project the hurricane impact based on the synthetic hurricane data set and the statistical power outage model (one logistic regression simplified from Nateghi et al. 2011). The results are shown in following table:

Model \ %	5 day historic power outage	5 day future power outage	5 day historic compound	5 day future compound
Physical	14	44	0.8	18
Statistical	13	38	0.7	16

The comparison shows that our power outage projection made by the physics-based model is comparable with statistical models at the census-tract levels: 1) the main conclusion that the compound hazard is going to be intensified in future is robust 2) the projections of power outage of physical and statistical model are similar on the total power outage level. However, note that the statistical model does not allow modification to network topology, it cannot reproduce the results shown in Figures 5 and 6. Nevertheless, we could test statistical model for Figure 3 to confirm that both approaches would get similar compound hazard estimations for both the current and future climates.

Though statistical model holds great prediction power and is easy to employ, it cannot be used to study strategies and policies that could be applied to the power systems. When the power system is modified, the whole prediction results would change. So when we are going to study retrofitting/ hardening policies for the power system to resist future hurricanes, the physics model might be the only choice. From Ref. 8, we believe the distribution grid is important, so we built and validated our model based on the distribution grid (synthetically though). Stand on this, we proved that for all the acyclic local distribution network structures (the topology of most distribution networks), the expected power outage under random attack would be proportional to the mean harmonic length of the distribution network branches, as stated in last response. This could be used to support the simplification on local distribution

network data. Also, as stated in the last response, we do not employ detailed level of power system down to the residence level. Our aim is to simulate the power failure and resilience processes at the census tract level and require the outage scenarios several days post-hurricanes. On this side, we believe our model could capture census tract level statistics (with validation results shown in last response and here; Fig. R1). (We have added references on the statistical models in Methods.)

Fig. R1. The initial power outage rate after hurricane Ike and Harvey under simulation comparing with observation on census tract level

Nateghi, R., Guikema, S. D., & Quiring, S. M. (2011). Comparison and validation of statistical methods for predicting power outage durations in the event of hurricanes. *Risk Analysis: An International Journal*, 31(12), 1897-1906.

- Specifically, the study on protective devices does not seem to be based on right scenarios. To protect the grid, power companies have installed all kind of protective devices, i.e., at the substations and transformers, not just those on the transmission lines. Therefore, it seems to be a big jump to claim that protective devices do not matter to failure patterns. Additionally, fault currents rather than overload often occur during weather events. Focusing on overload induced failures may be misleading for the distribution grid.

Thanks the reviewer again for questioning our simulation on protective devices. We notice that there exist many kinds of protective devices at both transmission and distribution systems, but as we responded last time, protective device data is generally not available [neither synthetic distribution grid papers for daily operation and for post-hazards mentioned about protective devices (Li et al. 2020 and Zhai et al. 2021)]. We did one experiment in the last response to simulate protective devices on the transmission line and find the settings do not change the main conclusion of this paper.

We also notice that in the presence of different protective devices, the system outage and recovery model could be very complicated, requiring not only capturing their trigger

mechanisms, but also modeling the self-recovery schemes and the re-energizing process using switching actions. But from related studies in the literature, repairing protective devices triggered outage can be very quick and usually at the second (with self-recovery scheme, Ji and Wei 2019) to minute or hourly scales (Arif et al., 2018; Chen et al., 2019), which seldomly affect the outage prediction at days level post-hurricanes in this research. A recent work by Tan, Qiu, Das, Kirschen et al. (2019) also did not consider the effect of protection devices during the post-disaster repairs in power distribution networks. The fault currents do occur in daily operations and weather related events, e.g. heavy rains, tornados. The fault currents will trigger protective/operational devices; however, they may not lead to structural damages that will harm the recovery process. We believe it is correct to use protective device data to simulate high-resolution outage and recovery at second or minute scale, but for our research purpose of estimating census-tract outage scenarios 5 days post-hurricanes, our current model is sufficient, as the results matched real outage and recovery data post-IKE and post-Harvey. (We have added these references to the Methods.)

Tan Y, Qiu F, Das AK, Kirschen DS, Arabshahi P, and Wang J. Scheduling Post-Disaster Repairs in Electricity Distribution Networks. IEEE Transactions on Power Systems. 2019, 34(4): 2611-2621.

Ji C and Wei Y. Dynamic Modeling and Resilience for Power Distribution. U.S. Patent, No . 62 / 065 , 408.

Arif A, Ma S, Wang Z, et al. Optimizing Service Restoration in Distribution Systems With Uncertain Repair Time and Demand. IEEE Transactions on Power Systems. 2018, 33(6): 6828-6838.

Chen B, Ye Z, Chen C and Wang J. Toward a MILP modeling framework for distribution system restoration. 2019, 34(3): 1749-1760.

--It is intuitive to consider climate change at the county level. However, aggregating over census track does not seem to solve the problem on what we do not know about failures. What key factors to include in the simulation may require more knowledge about how failures occurred at the grid.

Thanks the reviewer for questioning about our approach. We could feel the reviewer's deep interest in understanding how individual level power outage is happening. But as stated above, the post-hurricane outage and recovery modeling of the whole electric power system, including transmission system and distribution system, is very complicated, involving various processes and different device and operator actions. Constructing a high-resolution post-hurricane outage and recovery model for the whole power system is a huge ambition, and may make a big contribution to the field of electric power engineering. To make this clear, we have added to Method "The post-hurricane behavior of the electric power system, though simplified in this work to capture the census-level statistics of power outage, is extremely complex, involving numerous physical processes on the transmission and distribution networks and a variety of devices and operations. Future research may develop more detailed models to better capture physical mechanisms of post-hurricane power system failure." In this research, we do not aim to discover fundamental failure mechanisms, and we ignore some factors and many device/operator actions as our research only requires good census-tract-level prediction of

power outage at several days post-hurricanes to couple with the heat-wave model (at census-tract level) for the compound hazard risk analysis (involving 20 thousand simulations). After validating the current model using the real census-tract-level outage and recovery data post-IKE and post-Harvey, we believe the approach we are employing here is sufficient for the scope of our study and can support our main findings.

Reviewer 3#:

The authors have addressed my previous comments which are mostly related how they simulated the compound hazards. I do not have further questions.

Thank the reviewer for the positive comments.

Reviewers' Comments:

Reviewer #2:

Remarks to the Author:

The first part of this work is well done. Heatwave has happened after a server storm. Publishing the first part of the paper will be informative to the community.

For the second part of the paper, a main concern is not on physics based simulation but the approach itself. The model for the simulation misses several important aspects of physics on failures caused by extreme weather. Two examples are provided below.

(a) The paper does not have data on the major portion of the distribution grid. (Data are only on substations and road networks.) An understanding on failures also seems to be inadequate. For example, it is unclear from the paper and revision what failures are caused by hurricanes or heatwaves, and what the effects of protective devices are on outages. Without sufficient data or understanding, simulation does not seem to be well-grounded. Aggregation at census level does not help validation.

(b) Another example on a lack of understanding of physics is reflected by the view that the protective devices are unimportant. If one takes a step to understand more on the operational grid, it will become clear what role protective devices play. Please refer to the paper below also.

S. N. Talukdar et al. Cascading Failures: Survival versus Prevention.

https://research.ece.cmu.edu/cascadingfailures/Talukdar_CascadingFailuresSurvivalvsPrevention.pdf

Reviewer #3:

Remarks to the Author:

I do not have further questions and think this manuscript is ready for publication.

**Tropical cyclone-blackout-heatwave
Compound Hazard Resilience in a Changing Climate**

Kairui Feng, Ouyang Min, Ning Lin

Response to the reviewers

Line numbers refer to those in the revised manuscript

Reviewer #3 (Remarks to the Author):

I do not have further questions and think this manuscript is ready for publication.

Thanks to the reviewer for the positive comments. Thanks for checking on the details again.

Reviewer #2 (Remarks to the Author):

The first part of this work is well done. Heatwave has happened after a server storm. Publishing the first part of the paper will be informative to the community. For the second part of the paper, a main concern is not on physics based simulation but the approach itself. The model for the simulation misses several important aspects of physics on failures caused by extreme weather. Two examples are provided below.

Response: Thanks to the reviewer for acknowledging the first part of the paper informative. Let us first give a *Point-by-Point Response* to the reviewer's specific comments below, and then we will add an overall response to the reviewer's concern on the power outage and recovery modeling approach.

a) The paper does not have data on the major portion of the distribution grid. (Data are only on substations and road networks.)

Response: We fully understand the reviewer's concern on the synthetical data on the distribution network, making the findings seem not convincing. In fact, the distribution data is usually not available due to security concern. In the revision process, we contacted the CenterPoint Energy, and finally got the real distribution network data for Harris County. After calibrating the vulnerability and recovery parameters using the true distribution network data, we still found that the simulated restoration curve fits the real curve well, confirming the model useful again. This new result has been demonstrated in the revised main text at lines 525-528 "**Although the distribution network data is generally not available in a large domain, e.g., whole east coast of United States, the data is available for Harris County. The power system simulation results based on the true distribution networks are similarly accurate (Fig. S9d).**" We also performed the analysis on the risk mitigation part using the true distribution data, and we have added to Lines

569-574 “We test the effect of the assumption on the distribution networks by applying the analysis to the MST-based distribution networks and the true distribution networks and find similar results (Supplementary Fig. S10; e.g., for an enhancement rate of 5%, the expected percentage of residents experiencing at least one longer-than-5-day TC-blackout-compound hazard is different only by 2% between MST and star-like based models and by 3% between the true distribution networks and star-like based networks).”

Fig. S9. Impact of assumptions on the distribution network and protective devices. a) Post-Ike restoration curves averaged over 100 runs in the case of considering cascading failures and not considering cascading failures (i.e., not including protective devices, as they are mainly preventing the power systems from cascading failures), when the distribution networks are modelled as star-like. b) Fraction of overloaded lines during the restoration process when considering the overload-induced cascading failure under normal and reduced power demands (e.g., due to evacuation), for star-like distribution networks. c) Post-Ike restoration curves averaged over 100 runs in the case of considering cascading failures and not considering cascading failures when the distribution network are modelled as MST. d) Post-Ike restoration curves averaged over 100 runs under the true distribution networks (distribution network data obtained from CenterPoint Energy, Inc).

Fig. S10. Same as for Fig. 6, expect for MST-based distribution networks (a) and for the true distribution networks (b)

Currently there are no perfect distribution network data available for a large domain, so recent research often follow distribution network generation methods and match the statistical results with macro-observations. A recent work on synthetic distribution grid is done by Li et al. 2020 (led by NAE member Tomas Overbye). This is a large project sponsored by Department of Energy to develop synthetic power grid to be used as the real grid. However, they also don't have detailed distribution grid information other than statistics, so they matched their synthetic network metrics with real world metrics. The latest work to simulate power outage under natural hazards was conducted by Zhai et al. 2021 (led by Prof. Guikema from Umich, who is also an expert in post-hurricane power outage). Both Li et al. 2020 and Zhai et al. 2021 generated synthetic power systems assuming MST and/or star networks. Hence, in the main text, we are still reporting the main results we obtained with the synthetic distribution networks for more general interests (i.e., to illustrate that the method applies in general even when distribution network data is not available). But we have revised the description of the power modeling and analysis part in the Method to make this more clear (Lines 522-525), including **“To test the assumption on the distribution networks, we use the Harris road network to generate the MST structure of the power distribution networks and find that the specifics of the acyclic distribution network structure does not affect the accuracy of the global power system simulation (Figs. S9a and S9c).”**

Zhai, C., Chen, T. Y. J., White, A. G., & Guikema, S. D. (2021). Power outage prediction for natural hazards using synthetic power distribution systems. *Reliability Engineering & System Safety*, 208, 107348.

Li, H., Wert, J. L., Birchfield, A. B., Overbye, T. J., San Roman, T. G., Domingo, C. M., ... & Palmintier, B. (2020). Building highly detailed synthetic electric grid data sets for combined transmission and distribution systems. *IEEE Open Access Journal of Power and Energy*, 7, 478-488.

An understanding on failures also seems to be inadequate. For example, it is unclear from the paper and revision what failures are caused by hurricanes or heatwaves, and what the effects of protective devices are on outages. Without sufficient data or understanding, simulation does not seem to be well-grounded. Aggregation at census level does not help validation.

Response: The failures in this paper are caused by hurricane wind, as mentioned in Intro Lines 96-98 **“Here we focus on wind effects [on the power grid] (see Discussion) and apply a parametric model to estimate the spatial-temporal wind field for each synthetic storm.”** ([on the power grid] added in this revision to make it clear), in Discussion Lines 332-334 **“As the first attempt in quantifying the compound hazard risk, here we focus on the dominate power damage effects -- winds and induced falling trees”**, and in Method Lines 488-491, **“Given the TC wind hazard (i.e., local maximum wind gust during the storm), the power grid failure model first...”**.

The reviewer has a good point on the effect of heatwaves. Potentially, heatwave will impact daily operation and might damage power system (Wang et al. 2021), but we did not simulate these here as we are focusing on long-term post-hurricane power outage which are mainly caused by structural damage (an example shows below). We have added to Discussion Lines 346-348 **“The heatwave may also trigger power outages due to excessive power demand; however, these outages usually restore at a time scale of hours [49], rather than at a daily or weekly restore time scale for the TC damage induced outages that we consider in this study.”**

Hurricane Ida destroyed almost 30,000 utility poles. Some Louisiana residents could be without power for up to a month. Credit: Nick Wagner/Xinhua/eyevine (from Kozlov, Max. "Hurricane Ida forces Louisiana researchers to rethink their future." Nature (2021).)

Wang, Zhe, Tianzhen Hong, and Han Li. "Informing the planning of rotating power outages in heat waves through data analytics of connected smart thermostats for residential buildings." Environmental Research Letters 16, no. 7 (2021): 074003.

Our model is evaluated with two historical events (the only two in recent years for the study area) and various numerical experiments (i.e., different distribution network models and having not having protective devices, etc.) Census level results are sufficient for risk analysis under very large numbers of climate scenarios we focus on here. More discussion on the modeling approach is included below.

b) Another example on a lack of understanding of physics is reflected by the view that the protective devices are unimportant. If one takes a step to understand more on the operational grid, it will become clear what role protective devices play. Please refer to the paper below also.

S. N. Talukdar et al. Cascading Failures: Survival versus Prevention.

https://research.ece.cmu.edu/cascadingfailures/Talukdar_CascadingFailuresSurvivalvsPrevention.pdf

Response: We are sorry in the last response to make the reviewer misunderstood our arguments and added new analysis. For various cascading failure models in the literature, protective devices do play a significant role during the cascading failure process. However, we do not have the detailed location and type data of the protection devices, then as an alternative, we simply assume each transmission line has a protective device which cuts off the line once it is overloaded, and have investigated the impact of those settings on the outage and recovery process. Despite these experiments might still deviate from the reality, the experimental results are still insightful because the simulated results were well consistent with the real observation, illustrating that there exist other dominant factors influencing the power outage and recovery under hurricanes, i.e., the tree-like distribution network and a mass of physically damaged power system components. It is also worthy of note that repairing protective devices-triggered outage can be very quick and usually at the second (with self-recovery scheme, Ji and Wei 2019) to minute or hourly scales (Arif et al., 2018; Chen et al., 2019). The Northeast power outage at Aug. 14, 2003, as a typical example of protective device led cascading failures, lasted around 2 hours–4 days, depending on location, but most places restored power within 7 hours while the New York City Subway resumed limited services within 4 hours (CNN 2003). However, it took Houston 25 days to recover from Hurricane Ike. This is because hurricanes physically damage the power poles (it may also damage protective devices). And the repair time for each physically damaged power system component is much longer than just that for repairing the protective device. The damage figure above shows the typical damage situations under hurricane impact. This also means the physical damage of power system components dominates the power recovery process at the daily scale. As our research mainly discusses power outage and heatwave for longer than 5 days, the contribution of protective devices is not very significant considering our research purpose. This has been clarified in the revised manuscript at lines 511-514 **“Given that distribution networks and protective device data are generally not available, star-like network [35] and minimal spanning tree (MST) [73] models (and models combining these two [73,74]) are usually used to generate synthetic distribution networks and features that may have minor effects on predicting the daily scale power outage and recovery under hurricanes including protective devices are usually neglected [73-75].”**, Lines 528-541 **“Protective devices protect the distribution/transmission lines from overflow induced physical damages; however, the power outage caused by protective devices usually would be restored in hours, which is not at the same scale of the post-hurricane power outage (weekly level). To further investigate how the protective devices could potentially impact the power system resilience, we assume all power lines are equipped with protective devices and incorporated the overload-induced component failures into the power outage modeling (the same as in Ref. [75]). We find that the recovery process is similar between the case considering cascading failure and the case not considering cascading failure (prevented by protective devices) (Figs. S9a and S9c). Also, although under the normal demand there would**

be ~10% overloaded lines post the hurricane, when the demand is less than 95% of the normal demand (due to, e.g., hurricane evacuation) the overloading hardly happens (Fig. S9b). These findings illustrate that the impact of protective devices is relatively small on the daily-scale power outage and recovery under hurricanes, where a large number of physically damaged power system components often need days to weeks to fully repair.”

"Major power outage hits New York, other large cities". CNN. Turner Broadcasting System. August 14, 2003. Archived from the original on September 14, 2008. Retrieved September 16, 2008.

Ji C and Wei Y. Dynamic Modeling and Resilience for Power Distribution. U.S. Patent, No. 62 / 065 , 408.

Arif A, Ma S, Wang Z, et al. Optimizing Service Restoration in Distribution Systems With Uncertain Repair Time and Demand. IEEE Transactions on Power Systems. 2018, 33(6): 6828-6838.

Chen B, Ye Z, Chen C and Wang J. Toward a MILP modeling framework for distribution system restoration. 2019, 34(3): 1749-1760.

Thanks for pointing us to a reference related to protective device. A student of the authors (Hines, Paul) of the paper “Sarosh N. Talukdar, Jay Apt, Marija Ilic, Lester B. Lave, and M. Granger Morgan, Cascading Failures: Survival versus Prevention” who is now a professor in electrical engineering published another paper years after the publication of “Cascading Failures: Survival versus Prevention” that more clearly discussed about the simulation of protective device (Hines et al. 2010, our **Reference 75**). The description of the model proposed by Hines is copied as following for explanation: “After a component fails the model recalculates the power flow and advances to the time at which the next component fails or quits if no further components are overloaded. If a component failure separates the grid into unconnected sub-grids, the following process is used to re-balance supply and demand. If the imbalance is small, such that generators can adjust their output by not more than 10% and arrive at a new supply/demand balance, this balance is achieved through generator set-point adjustments. If this adjustment is insufficient, the smallest generator in the sub-grid is shut down until there is an excess of load. If there is excess load after these generator adjustments, the simulator curtails enough load to balance supply and demand”.

The difference between the Hines’ model and our adapted model lies in the treatment of overload-induced component failures in each sub-grid after the initial failures. To test the impact, we took this into account (with more computational cost) and updated the power response model (to be the same as the Hines’ model), and the results are almost unchanged. We further investigated how much this additional consideration of “overload- induced component failures” affects the estimation of the number of customers with power (**Fig. S9a-9c**), as discussed above.

Hines, Paul, Eduardo Cotilla-Sanchez, and Seth Blumsack. "Do topological models provide good information about electricity infrastructure vulnerability?." *Chaos: An Interdisciplinary Journal of Nonlinear Science* 20.39 (2010): 033122.

Overall Response to reviewer's concern on the power system modeling approach: we acknowledge that the power outage mechanism that involves the cascading failure process is very complicated. This makes dozens of cascading failure models for power systems being proposed in the literature, including the Manchester AC Model (Rios et al. 2002; Kirschen et al. 2004), improved Manchester model (Mei et al. 2008), the OPA model (Dobson et al. 2001; Carreras et al. 2004), the improved OPA model (Mei et al. 2009), the hidden failure models (Wang and Thorp 2001), an analytical CASCADE model (Dobson et al. 2005), an branch process model (Ren and Dobson 2008), Hines' model (Hines et al. 2010), a stochastic model (Anghel et al 2007), and many other models inspired by network science (Motter and Lai 2002;), and also a huge number of models considering the failures due to interdependencies with other infrastructure systems (Zeng et al. 2019; He et al. 2018). Many of these models were proposed by experts in the field of power engineering and published in authoritative journals including IEEE Transactions on Power Systems. Despite significant differences of these models, they can well reproduce different observations during or after the cascading failures (Dobson et al. 2016). If simply judging these models in comparison to the real cascading failure process, they might be all wrong, but they are still useful in terms of whether the observation could be well reproduced. Within this context, we acknowledge that the power outage model in this work is incapable to simulate all details on failure mechanisms of power systems under extreme weather, but it can well predict the daily-scale power outage and recovery according to the validation analysis using historical observations. Also, the census-tract level outage scenarios 5 days post-hurricanes are the main outputs we expect from the model for further analysis, so from this aspect of observation we believe the model is useful enough to support our research as well as the main findings. This has been clarified in the revised manuscript at lines 516-520 **“The well consistency between the daily scale simulated results using this model and the real observations, together with the fact that this research mainly takes its produced census-tract level outage scenarios to support the compound hazard resilience analysis, proves the power system model useful enough to support this research as well as the main findings.”** And Lines 541-546 **“However, it is still worthy of note that the post-hurricane behavior of the electric power system, though simplified in this work to capture the census-level statistics of power outage, is extremely complex, involving numerous physical processes on the transmission and distribution networks and a variety of devices and operations. Future research may develop more detailed models to better capture physical mechanisms of post-hurricane power system failure.”**

- [1] Kirschen D S, Jayaweera D, Nedic D P, et al. A probabilistic indicator of system stress. IEEE Transactions on Power Systems, 2004, 19(3): 1650-1657.
- [2] Rios M A, Kirschen D S, Jayaweera D, et al. Value of security: modeling time-dependent phenomena and weather conditions. IEEE Transactions on Power Systems, 2002, 17(3): 543-548.
- [3] Dobson I, Carreras BA, Lynch VE and Newman DE. An initial model for complex dynamics

- in electric power system blackouts. Hawaii International Conference on System Sciences, January 2001, Maui, Hawaii.
- [4] Carreras B A, Newman D E, Dobson I, et al. Evidence for self-organized criticality in a time series of electric power system blackouts. *IEEE Transactions on Circuits and Systems I: Regular Papers*, 2004, 51(9): 1733-1740.
 - [5] Dobson I, Carreras B A, Newman D E, et al. Obtaining statistics of cascading line outages spreading in an electric transmission network from standard utility data. *IEEE Transactions on Power Systems*, 2016, 31(6): 4831-4841.
 - [6] Dobson I., Carreras B.A. and Newman D.E. (2005). A loading-dependent model of probabilistic cascading failure. *Probability in engineering and informational science*, 19, 15-32.
 - [7] Hines, Paul, Eduardo Cotilla-Sanchez, and Seth Blumsack. "Do topological models provide good information about electricity infrastructure vulnerability?." *Chaos: An Interdisciplinary Journal of Nonlinear Science* 20.39 (2010): 033122.
 - [8] Ren H, Dobson I. Using transmission line outage data to estimate cascading failure propagation in an electric power system. *IEEE Transactions on Circuits and Systems II: Express Briefs*, 2008, 55(9): 927-931.
 - [9] Anghel M, Werley KA and Motter AE. Stochastic model for power grid dynamics. Fortieth Hawaii International Conference on System Sciences, January 3-6, 2007, Big Island, Hawaii.
 - [10] Motter AE and Lai YC. Cascade-based attacks on complex networks. *PHYSICAL REVIEW E*, 2002, 66, 065102.
 - [11] Wang H, Thorp J S. Optimal locations for protection system enhancement: a simulation of cascading outages. *IEEE Transactions on Power Delivery*, 2001, 16(4): 528-533.
 - [12] Mei S, Ni Y, Wang G, et al. A study of self-organized criticality of power system under cascading failures based on AC-OPF with voltage stability margin. *IEEE Transactions on Power Systems*, 2008, 23(4): 1719-1726.
 - [13] Mei S, He F, Zhang X, et al. An improved OPA model and blackout risk assessment. *IEEE Transactions on Power Systems*, 2009, 24(2): 814-823.
 - [14] Zeng Z, Ding T, Xu Y, et al. Reliability evaluation for integrated power-gas systems with power-to-gas and gas storages. *IEEE Transactions on Power Systems*, 2019, 35(1): 571-583.
 - [15] He C, Dai C, Wu L, et al. Robust network hardening strategy for enhancing resilience of integrated electricity and natural gas distribution systems against natural disasters. *IEEE Transactions on Power Systems*, 2018, 33(5): 5787-5798.

Reviewers' Comments:

Reviewer #2:

Remarks to the Author:

The reviewer has no further questions on the first part of the paper on the compound hazard of tropical cyclone and heatwave. This part is well developed.

However, the second part of the paper has little improvement on a convincing validation of simulations involving the power grid. A new component from this revision is that data is claimed to be obtained from the grid in the area under study. However, no descriptions are given on the data itself. From Fig. S9, it

seems that the data is on the percentages of customers who had or did not have power outages. If so, such data does not help validate simulations involving the power grid. The reason is that such data is too aggregated. Many models or simulations may have similar aggregated behavior. In general, validating a physics-based model at a crude aggregation level does not seem to be rigorous nor convincing when coupled with a climate model.

Reviewer #4:

Remarks to the Author:

This is a good paper looking at electric grids risks due to extreme weather with Houston area as an example.

Tropical cyclone-blackout-heatwave compound hazard resilience in a changing climate

Kairui Feng, Ouyang Min, Ning Lin

Response to the reviewers

Line numbers refer to those in the revised manuscript

Reviewer #2 (Remarks to the Author):

The reviewer has no further questions on the first part of the paper on the compound hazard of tropical cyclone and heatwave. This part is well developed. However, the second part of the paper has little improvement on a convincing validation of simulations involving the power grid. A new component from this revision is that data is claimed to be obtained from the grid in the area under study. However, no descriptions are given on the data itself. From Fig. S9, it seems that the data is on the percentages of customers who had or did not have power outages. If so, such data does not help validate simulations involving the power grid. The reason is that such data is too aggregated. Many models or simulations may have similar aggregated behavior. In general, validating a physics-based model at a crude aggregation level does not seem to be rigorous nor convincing when coupled with a climate model.

Response: Thanks for reviewing our manuscript again. We used true topography data of the power grid to evaluate the model and the results shows the model can produce similarly good results for the synthetic grid and the true grid, compared to the observed power outage data. We acquired the data from CenterPoint Energy, Inc. The data we obtained from CenterPoint Energy, Inc. are topological data, which includes the distribution network's topology along streets. And, to conduct the simulation, we assume that each household is linked to the nearest distribution network node. As such, it is a finer level of data than the synthetic network data. We've edited the text as "Although the distribution network **topology** is generally not available in a large domain, e.g., whole east coast of United States, the data is available for Harris County." to clarify we are using the topology network data. We have also added "**distribution network topology data obtained from CenterPoint Energy, Inc**" to the figure caption of Figure S9, where the comparison is shown. Also, we updated the data availability statement as "**The power network data (including topology data for transmission and distribution networks) and the historical power outage data were obtained from CenterPoint Energy, Inc.**"

We show only the validation of aggregated level power outage over Harris County in Fig. S9, as we want to show that our framework could predict power outage accurately based on the true power network data. However, **Fig. S2** (shown below) shows the comparison of the simulated and observed initial power outage rate for Hurricanes Ike and Harvey **at the census tract level**, which shows the prediction from our model matches with the census tract level power outage observations well. The resolution of census tract would be useful to support decision making for policy makers.

Fig. S2. Comparison of the simulated and observed initial power outage rate for Hurricanes Ike and Harvey at the census tract level (using star-like network topology).

Reviewer #4 (Remarks to the Author):

This is a good paper looking at electric grids risks due to extreme weather with Houston area as an example.

Thanks to the reviewer for the positive comment.